# LDReg: Local Dimensionality Regularized Self-Supervised Learning

**Hanxun Huang**[1]  **Ricardo J. G. B. Campello**[2]  **Sarah Erfani**[1]  **Xingjun Ma**[3]
**Michael E. Houle**[4]  **James Bailey**[1]

[1]School of Computing and Information Systems, The University of Melbourne, Australia
[2]Department of Mathematics and Computer Science, University of Southern Denmark, Denmark
[3]School of Computer Science, Fudan University, China
[4]Department of Computer Science, New Jersey Institute of Technology, USA

## ABSTRACT

Representations learned via self-supervised learning (SSL) can be susceptible to dimensional collapse, where the learned representation subspace is of extremely low dimensionality and thus fails to represent the full data distribution and modalities. Dimensional collapse —— also known as the "underfilling" phenomenon —— is one of the major causes of degraded performance on downstream tasks. Previous work has investigated the dimensional collapse problem of SSL at a global level. In this paper, we demonstrate that representations can span over high dimensional space globally, but collapse locally. To address this, we propose a method called *local dimensionality regularization (LDReg)*. Our formulation is based on the derivation of the Fisher-Rao metric to compare and optimize local distance distributions at an asymptotically small radius for each data point. By increasing the local intrinsic dimensionality, we demonstrate through a range of experiments that LDReg improves the representation quality of SSL. The results also show that LDReg can regularize dimensionality at both local and global levels.

## 1 INTRODUCTION

Self-supervised learning (SSL) is now approaching the same level of performance as supervised learning on numerous tasks (Chen et al., 2020a;b; He et al., 2020; Grill et al., 2020; Chen & He, 2021; Caron et al., 2021; Zbontar et al., 2021; Chen et al., 2021; Bardes et al., 2022; Zhang et al., 2022). SSL focuses on the construction of effective representations without reliance on labels. Quality measures for such representations are crucial to assess and regularize the learning process. A key aspect of representation quality is to avoid dimensional collapse and its more severe form, mode collapse, where the representation converges to a trivial vector (Jing et al., 2022). Dimensional collapse refers to the phenomenon whereby many of the features are highly correlated and thus span only a lower-dimensional subspace. Existing works have connected dimensional collapse with low quality of learned representations (He & Ozay, 2022; Li et al., 2022; Garrido et al., 2023a; Dubois et al., 2022). Both contrastive and non-contrastive learning can be susceptible to dimensional collapse (Tian et al., 2021; Jing et al., 2022; Zhang et al., 2022), which can be mitigated by regularizing dimensionality as a global property, such as learning decorrelated features (Hua et al., 2021) or minimizing the off-diagonal terms of the covariance matrix (Zbontar et al., 2021; Bardes et al., 2022).

In this paper, we examine an alternative approach to the problem of dimensional collapse, by investigating the local properties of the representation. Rather than directly optimizing the global dimensionality of the entire training dataset (in terms of correlation measures), we propose to regularize the local intrinsic dimensionality (LID) (Houle, 2017a;b) at each training sample. We provide an intuitive illustration of the idea of LID in Figure 1. Given a representation vector (anchor point) and its surrounding neighbors, if representations collapse to a low-dimensional space, it would result in a lower sample LID for the anchor point (Figure 1a). In SSL, each anchor point should be dissimilar from all other points and should have a higher sample LID (Figure 1b). Based on LID, we reveal an interesting observation that: *representations can span a high dimensional space globally, but collapse locally*. As shown in the top 4 subfigures in Figure 1c, the data points could span

over different local dimensional subspaces (LIDs) while having roughly the same global intrinsic dimension (GID). This suggests that dimensional collapse should not only be examined as a global property but also locally. Note that Figure 1c illustrates a synthetic case of local dimensional collapse. Later we will empirically show that representations converging to a locally low-dimensional subspace can have reduced quality and that higher LID is desirable for SSL.

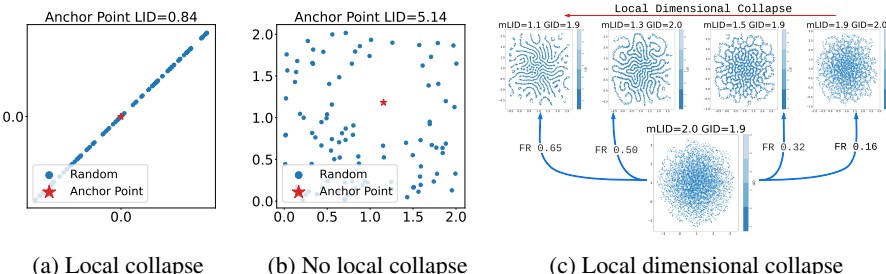

(a) Local collapse      (b) No local collapse      (c) Local dimensional collapse

Figure 1: Illustrations with 2D synthetic data. (a-b) The LID value of the anchor point (red star) when there is (or is no) local collapse. (c) Fisher-Rao (FR) metric and mean LID (mLID) estimates. FR measures the distance between two LID distributions, and is computed based on our theoretical results. mLID is the geometric mean of sample-wise LID scores. High FR distances and low mLID scores indicate greater dimensional collapse. Global intrinsic dimension (GID) is estimated using the DanCO algorithm (Ceruti et al., 2014).

To address local dimensional collapse, we propose the *Local Dimensionality Regularizer (LDReg)*, which regularizes the representations toward a desired local intrinsic dimensionality to avoid collapse, as shown at the bottom subfigure of Figure 1c. Our approach leverages the LID Representation Theorem (Houle, 2017a), which has established that the distance distribution of nearest neighbors in an asymptotically small radius around a given sample is guaranteed to have a parametric form. For LDReg to be able to influence the learned representations toward a *distributionally higher* LID, we require a way to compare distributions that is sensitive to differences in LID. This motivates us to develop a new theory to enable measurement of the 'distance' between local distance distributions, as well as identify the mean of a set of local distance distributions. We derive a theoretically well-founded Fisher-Rao metric (FR), which considers a statistical manifold for assessing the distance between two local distance distributions in the asymptotic limit. As shown in Figure 1c, FR corresponds well with different degrees of dimensional collapse. More details regarding Figure 1c can be found in Appendix A.

The theory we develop here also leads to two new insights: **i)** LID values are better compared using the logarithmic scale rather than the linear scale; **ii)** For aggregating LID values, the geometric mean is a more natural choice than the arithmetic or harmonic means. These insights have consequences for formulating our local dimensionality regularization objective, as well as broader implications for comparing and reporting LID values in other contexts.

To summarize, the main contributions of this paper are:

- A new approach, LDReg, for mitigating dimensional collapse in SSL via the regularization of local intrinsic dimensionality characteristics.
- Theory to support the formulation of LID regularization, insights into how dimensionalities should be compared and aggregated, and generic dimensionality regularization technique that can potentially be used in other types of learning tasks.
- Consistent empirical results demonstrating the benefit of LDReg in improving multiple state-of-the-art SSL methods (including SimCLR, SimCLR-Tuned, BYOL, and MAE), and its effectiveness in addressing both local and global dimensional collapse.

## 2 RELATED WORK

**Self-Supervised Learning (SSL).** SSL aims to automatically learn high-quality representations without label supervision. Existing SSL methods can be categorized into two types: generative

methods and contrastive methods. In generative methods, the model learns representations through a reconstruction of the input (Hinton & Zemel, 1993). Inspired by masked language modeling (Kenton & Toutanova, 2019), recent works have successfully extended this paradigm to the reconstruction of masked images (Bao et al., 2022; Xie et al., 2022), such as Masked AutoEncoder (MAE) (He et al., 2022). It has been theoretically proven that these methods are a special form of contrastive learning that implicitly aligns positive pairs (Zhang et al., 2022). Contrastive methods can further be divided into 1) sample-contrastive, 2) dimension-contrastive (Garrido et al., 2023b), and 3) asymmetrical models. SimCLR (Chen et al., 2020a) and other sample-contrastive methods (He et al., 2020; Chen et al., 2020a;b; Yeh et al., 2022) are based on InfoNCE loss (Oord et al., 2018). The sample-contrastive approach has been extended by using nearest-neighbor methods (Dwibedi et al., 2021; Ge et al., 2023), clustering-based methods (Caron et al., 2018; 2020; Pang et al., 2022), and improved augmentation strategies (Wang et al., 2023). Dimension-contrastive methods (Zbontar et al., 2021; Bardes et al., 2022) regularize the off-diagonal terms of the covariance matrix of the embedding. Asymmetrical models use an asymmetric architecture, such as an additional predictor (Chen & He, 2021), self-distillation (Caron et al., 2021), or a slow-moving average branch as in BYOL (Grill et al., 2020).

**Dimensional collapse in SSL.** Dimensional collapse occurs during the SSL process where the learned embedding vectors and representations span only a lower-dimensional subspace (Hua et al., 2021; Jing et al., 2022; He & Ozay, 2022; Li et al., 2022). Generative methods such as MAE (He et al., 2022) have been shown to be susceptible to dimensional collapse (Zhang et al., 2022). Sample-contrastive methods such as SimCLR have also been observed to suffer from dimensional collapse (Jing et al., 2022). Other studies suggest that while stronger augmentation and larger projectors are beneficial to the performance (Garrido et al., 2023b), they may cause a dimensional collapse in the projector space (Cosentino et al., 2022). It has been theoretically proven that asymmetrical model methods can alleviate dimensional collapse, and the effective rank (Roy & Vetterli, 2007) is a useful measure of the degree of global collapse (Zhuo et al., 2023). Effective rank is also helpful in assessing the representation quality (Garrido et al., 2023a). By decorrelating features, dimension-contrastive methods (Zbontar et al., 2021; Zhang et al., 2021; Ermolov et al., 2021; Bardes et al., 2022) can also avoid dimensional collapse. In this work, we focus on the local dimensionality of the representation (encoder) space, which largely determines the performance of downstream tasks.

**Local Intrinsic Dimensionality.** Unlike global intrinsic dimensionality metrics (Pettis et al., 1979; Bruske & Sommer, 1998), local intrinsic dimensionality (LID) measures the intrinsic dimension in the vicinity of a particular query point (Levina & Bickel, 2004; Houle, 2017a). It has been used as a measure for similarity search (Houle et al., 2012), for characterizing adversarial subspaces (Ma et al., 2018a), for detecting backdoor attacks (Dolatabadi et al., 2022), and in the understanding of deep learning (Ma et al., 2018b; Gong et al., 2019; Ansuini et al., 2019; Pope et al., 2021). In Appendix B, we provide a comparison between the effective rank and the LID to help understand local vs. global dimensionality. Our work in this paper shows that LID is not only useful as a descriptive measure, but can also be used as part of a powerful regularizer for SSL.

## 3 BACKGROUND AND TERMINOLOGY

We first introduce the necessary background for the distributional theory underpinning LID. The dimensionality of the local data submanifold in the vicinity of a reference sample is revealed by the growth characteristics of the cumulative distribution function of the local distance distribution.

Let $F$ be a real-valued function that is non-zero over some open interval containing $r \in \mathbb{R}$, $r \neq 0$.

**Definition 1** ((Houle, 2017a))**.** *The* intrinsic dimensionality *of $F$ at $r$ is defined as follows, whenever the limit exists:*

$$\mathrm{IntrDim}_F(r) \quad \triangleq \quad \lim_{\epsilon \to 0} \frac{\ln\left(F((1+\epsilon)r)/F(r)\right)}{\ln((1+\epsilon)r/r)}.$$

**Theorem 1** ((Houle, 2017a))**.** *If $F$ is continuously differentiable at $r$, then*

$$\mathrm{LID}_F(r) \quad \triangleq \quad \frac{r \cdot F'(r)}{F(r)} \quad = \quad \mathrm{IntrDim}_F(r).$$

Although the preceding definitions apply more generally, we will be particularly interested in functions $F$ that satisfy the conditions of a cumulative distribution function (CDF). Let **x** be a location of

interest within a data domain $\mathcal{S}$ for which the distance measure $d : \mathcal{S} \times \mathcal{S} \to \mathbb{R}_{\geq 0}$ has been defined. To any generated sample $\mathbf{s} \in \mathcal{S}$, we associate the distance $d(\mathbf{x}, \mathbf{s})$; in this way, a *global* distribution that produces the sample $\mathbf{s}$ can be said to induce the random value $d(\mathbf{x}, \mathbf{s})$ from a *local* distribution of distances taken with respect to $\mathbf{x}$. The CDF $F(r)$ of the local distance distribution is simply the probability of the sample distance lying within a threshold $r$ — that is, $F(r) \triangleq \Pr[d(\mathbf{x}, \mathbf{s}) \leq r]$.

To characterize the local intrinsic dimensionality in the vicinity of location $\mathbf{x}$, we consider the limit of $\mathrm{LID}_F(r)$ as the distance $r$ tends to $0$. Regardless of whether $F$ satisfies the conditions of a CDF, we denote this limit by

$$\mathrm{LID}_F^* \triangleq \lim_{r \to 0^+} \mathrm{LID}_F(r).$$

Henceforth, when we refer to the local intrinsic dimensionality (LID) of a function $F$, or of a point $\mathbf{x}$ whose induced distance distribution has $F$ as its CDF, we will take 'LID' to mean the quantity $\mathrm{LID}_F^*$.

In general, $\mathrm{LID}_F^*$ is not necessarily an integer. Unlike the manifold model of local data distributions — where the dimensionality of the manifold is always an integer, and deviation from the manifold is considered as 'error' — the LID model reflects the entire local distributional characteristics without distinguishing error. However, the estimation of the LID at $\mathbf{x}$ often gives an indication of the dimension of the local manifold containing $\mathbf{x}$ that would best fit the distribution.

## 4 ASYMPTOTIC FORM OF FISHER-RAO METRIC FOR LID DISTRIBUTIONS

We now provide the necessary theoretical justifications for our LDReg regularizer which will be later developed in Section 5. Intuitively, LDReg should regularize the LID of the local distribution of the training samples towards a higher value, determined as the LID of some target distribution. This can help to avoid dimensional collapse by increasing the dimensionality of the representation space and producing representations that are more uniform in their local dimensional characteristics. To achieve this, we will need an asymptotic notion of distributional distance that applies to lower tail distributions. In this section, we introduce an asymptotic variant of the Fisher-Rao distance that can be used to identify the center (mean) of a collection of tail distributions.

### 4.1 FISHER-RAO DISTANCE METRIC

The Fisher-Rao distance is based on the embedding of the distributions on a Riemannian manifold, where it corresponds to the length of the geodesic along the manifold between the two distributions. The metric is usually impossible to compute analytically, except for special cases (such as certain varieties of Gaussians). However, in the asymptotic limit as $w \to 0$, we will show that it is analytically tractable for smooth growth functions.

**Definition 2.** *Given a non-empty set $\mathcal{X}$ and a family of probability density functions $\phi(\mathbf{x}|\boldsymbol{\theta})$ parameterized by $\boldsymbol{\theta}$ on $\mathcal{X}$, the space $\mathcal{M} = \{\phi(\mathbf{x}|\boldsymbol{\theta})|\boldsymbol{\theta} \in \mathbb{R}^d\}$ forms a Riemannian manifold. The Fisher-Rao Riemannian metric on $\mathcal{M}$ is a function of $\boldsymbol{\theta}$ and induces geodesics, i.e., curves with minimum length on $\mathcal{M}$. The Fisher-Rao distance between two models $\boldsymbol{\theta_1}$ and $\boldsymbol{\theta_2}$ is the arc-length of the geodesic that connects these two points.*

In our context, we will focus on univariate lower tail distributions with a single parameter $\theta$ corresponding to the LID of the CDF. In this context, the Fisher-Rao distance will turn out to have an elegant analytical form. We will make use of the Fisher information $\mathcal{I}$, which is the variance of the gradient of the log-likelihood function (also known as the Fisher score). For distributions over $[0, w]$ with a single parameter $\theta$, this is defined as

$$\mathcal{I}_w(\theta) = \int_0^w \left( \frac{\partial}{\partial \theta} \ln F_w'(r|\theta) \right)^2 F_w'(r|\theta) \, \mathrm{d}r.$$

**Lemma 1.** *Consider the family of tail distributions on $[0, w]$ parameterized by $\theta$, whose CDFs are smooth growth functions of the form*

$$H_{w|\theta}(r) = \left( \frac{r}{w} \right)^{\theta}.$$

The Fisher-Rao distance $d_{\mathrm{FR}}$ between $H_{w|\theta_1}$ and $H_{w|\theta_2}$ is

$$d_{\mathrm{FR}}(H_{w|\theta_1}, H_{w|\theta_2}) = \left| \ln \frac{\theta_2}{\theta_1} \right| \ .$$

The Fisher information $\mathcal{I}_w$ for smooth growth functions of the form $H_{w|\theta}$ is:

$$\mathcal{I}_w(\theta) \ = \ \int_0^w \left( \frac{\partial}{\partial \theta} \ln H'_{w|\theta}(r) \right)^2 H'_{w|\theta}(r) \, \mathrm{d}r \ = \ \frac{1}{\theta^2} \ .$$

The proof of Lemma 1 can be found in Appendix C.2.

## 4.2 ASYMPTOTIC FISHER-RAO METRIC

We now extend the notion of the Fisher-Rao metric to distance distributions whose CDFs (conditioned to the lower tail $[0, w]$) have the more general form of a growth function. The LID Representation Theorem (Theorem 3 in Appendix C.1) tells us that any such CDF $F_w(r)$ can be decomposed into the product of a canonical form $H_{w|\,\mathrm{LID}_F^*}(r)$ with an auxiliary factor $A_F(r, w)$:

$$F_w(r) \ = \ H_{w|\,\mathrm{LID}_F^*}(r) \cdot A_F(r, w) \ = \ \left( \frac{r}{w} \right)^{\mathrm{LID}_F^*} \cdot \exp \left( \int_r^w \frac{\mathrm{LID}_F^* - \mathrm{LID}_F(t)}{t} \, \mathrm{d}t \right) \ .$$

From Corollary 3.1 (Appendix C.1), the auxiliary factor $A_F(r, w)$ tends to 1 as $r$ and $w$ tend to 0, provided that $r$ stays within a constant factor of $w$. Asymptotically, then, $F_w$ can be seen to tend to $H_{w|\theta}$ as the tail length tends to zero, for $\theta = \mathrm{LID}_F^*$. More precisely, for any constant $c \geq 1$,

$$\lim_{\substack{w \to 0^+ \\ w/c \leq r \leq cw}} \frac{F_w(r)}{H_{w|\,\mathrm{LID}_F^*}(r)} \ = \ \lim_{\substack{w \to 0^+ \\ w/c \leq r \leq cw}} A_F(r, w) \ = \ 1 \ .$$

Thus, although the CDF $F_w$ does not in general admit a finite parameterization suitable for the direct definition of a Fisher-Rao distance, asymptotically it tends to a distribution that does: $H_{w|\,\mathrm{LID}_F^*}$.

Using Lemma 1 we define an asymptotic form of Fisher-Rao distance between distance distributions.

**Definition 3.** *Given two smooth-growth distance distributions with CDFs $F$ and $G$, their* asymptotic Fisher-Rao distance *is given by*

$$d_{\mathrm{AFR}}(F, G) \ \triangleq \ \lim_{w \to 0^+} d_{\mathrm{FR}}(H_{w|\,\mathrm{LID}_F^*}, H_{w|\,\mathrm{LID}_G^*}) \ = \ \left| \ln \frac{\mathrm{LID}_G^*}{\mathrm{LID}_F^*} \right| \ .$$

## 4.3 IMPLICATIONS

**Remark 1.1.** *Assume that $\mathrm{LID}_F^* \geq 1$ and that $G_w = \mathcal{U}_{1,w}$ is the one-dimensional uniform distribution over the interval $[0, w]$ (with $\mathrm{LID}_G^*$ therefore equal to 1). We then have*

$$\begin{aligned} d_{\mathrm{AFR}}(F_w, \mathcal{U}_{1,w}) \ &= \ \ln \mathrm{LID}_F^* \\ \mathrm{LID}_F^* \ &= \ \exp \left( d_{\mathrm{AFR}}(F_w, \mathcal{U}_{1,w}) \right) \ . \end{aligned}$$

*We can therefore interpret the local intrinsic dimensionality of a distribution $F$ conditioned to the interval $[0, w]$ (with $\mathrm{LID}_F^* \geq 1$) as the exponential of the distance between distribution $F$ and the uniform distribution in the limit as $w \to 0$.*

There is also a close relationship between our asymptotic Fisher-Rao distance metric and a mathematically special measure of relative difference.

**Remark 1.2.** *One can interpret the quantity $\left| \ln \left( \mathrm{LID}_G^* / \mathrm{LID}_F^* \right) \right|$ as a relative difference between $\mathrm{LID}_G^*$ and $\mathrm{LID}_F^*$. Furthermore, it is the only measure of relative difference that is both symmetric, additive, and normed (Törnqvist et al., 1985).*

The asymptotic Fisher-Rao distance indicates that the absolute difference between the LID values of two distance distributions is not a good measure of asymptotic dissimilarity. For example, a

pair of distributions with $\text{LID}^*_{F^1} = 2$ and $\text{LID}^*_{G^1} = 4$ are much less similar under the asymptotic Fisher-Rao metric than a pair of distributions with $\text{LID}^*_{F^2} = 20$ and $\text{LID}^*_{G^2} = 22$.

We can also use the asymptotic Fisher-Rao metric to compute the 'centroid' or Fréchet mean[1] of a set of distance distributions, as well as the associated Fréchet variance.

**Definition 4.** *Given a set of distance distribution CDFs* $\mathcal{F} = \{F^1, F^2, \ldots, F^N\}$*, the empirical Fréchet mean of* $\mathcal{F}$ *is defined as*

$$\mu_{\mathcal{F}} \triangleq \underset{H_{w|\theta}}{\arg\min} \frac{1}{N} \sum_{i=1}^{N} \left( d_{\text{AFR}}(H_{w|\theta}, F^i) \right)^2 .$$

*The Fréchet variance of* $\mathcal{F}$ *is then defined as*

$$\sigma^2_{\mathcal{F}} \triangleq \frac{1}{N} \sum_{i=1}^{N} \left( d_{\text{AFR}}(\mu_{\mathcal{F}}, F^i) \right)^2 = \frac{1}{N} \sum_{i=1}^{N} \left( \ln \text{LID}^*_{F^i} - \ln \text{LID}^*_{\mu_{\mathcal{F}}} \right)^2 .$$

The Fréchet variance can be interpreted as the variance of the local intrinsic dimensionalities of the distributions in $\mathcal{F}$, taken in logarithmic scale.

The Fréchet mean has a well-known close connection to the geometric mean, when the distance is expressed as a difference of logarithmic values. For our setting, we state this relationship in the following theorem, the proof of which can be found in Appendix C.3.

**Theorem 2.** *Let* $\mu_{\mathcal{F}}$ *be the empirical Fréchet mean of a set of distance distribution CDFs* $\mathcal{F} = \{F^1, F^2, \ldots, F^N\}$ *using the asymptotic Fisher-Rao metric* $d_{\text{AFR}}$*. Then* $\text{LID}^*_{\mu_{\mathcal{F}}} = \exp\left( \frac{1}{N} \sum_{i=1}^{N} \ln \text{LID}^*_{F^i} \right)$*, the geometric mean of* $\{\text{LID}^*_{F^1}, \ldots, \text{LID}^*_{F^N}\}$*.*

**Corollary 2.1.** *Given the CDFs* $\mathcal{F} = \{F^1, F^2, \ldots, F^N\}$*, the quantity* $\frac{1}{N} \sum_{i=1}^{N} \ln \text{LID}^*_{F^i}$ *is:*

1. *The average asymptotic Fisher-Rao distance of members of* $\mathcal{F}$ *to the one-dimensional uniform distribution (if for all* $i$ *we have* $\text{LID}^*_{F^i} \geq 1$*).*

2. *The logarithm of the local intrinsic dimension of the Fréchet mean of* $\mathcal{F}$*.*

3. *The logarithm of the geometric mean of the local intrinsic dimensions of the members of* $\mathcal{F}$*.*

The proof of Assertion 1 is in Appendix C.4. Assertions 2 and 3 follow from Theorem 2.

It is natural to consider whether other measures of distributional divergence could be used in place of the asymptotic Fisher-Rao metric in the derivation of the Fréchet mean. Bailey et al. (2022) have shown several other divergences involving the LID of distance distributions — most notably that of the Kullback-Leibler (KL) divergence. We can in fact show that the asymptotic Fisher-Rao metric is preferable (in theory) to the asymptotic KL distance, and the geometric mean is preferable to the arithmetic mean and harmonic mean when aggregating the LID values of distance distributions. For the details, we refer readers to Theorem 4 in Appendix C.5.

In summary, Theorems 2 and 4 (Appendix C.5) show that the asymptotic Fisher-Rao metric is preferable in measuring the distribution divergence of LIDs. These theorems provide theoretical justification for LDReg, which will be described in the following section.

## 5 LID Regularization for Self-Supervised Learning

In this section, we formally introduce our proposed LDReg method. For an input image $\mathbf{x}$ and an encoder $f(\cdot)$, the representation of $\mathbf{x}$ can be obtained as $\mathbf{z} = f(\mathbf{x})$. Depending on the SSL method, a projector $g(\cdot)$, a predictor $h(\cdot)$, and a decoder $t(\cdot)$ can be used to obtain the embedding vector or the reconstructed image from $\mathbf{z}$. LDReg is a generic regularization on representations $\mathbf{z}$ obtained by the encoder, and as such it can be applied to a variety of SSL methods (more details are in Appendix E). We denote the objective function of an SSL method by $\mathcal{L}^{\text{SSL}}$.

---

[1] Also known as the Karcher mean, the Riemannian barycenter and the Riemannian center of mass.

**Local Dimensionality Regularization (LDReg).** Following theorems derived in Section 4, we assume that the representational dimension is $d$, and that we are given the representation of a sample $\mathbf{x}_i$. Suppose that the distance distribution induced by $F$ at $\mathbf{x}_i$ is $F_w^i(r)$. To avoid dimensional collapse, we consider maximizing the distributional distance between $F_w^i(r)$ and a uniform distance distribution $\mathcal{U}_{1,w}(r)$ (with LID $= 1$): for each sample, we could regularize a local representation that has a local intrinsic dimensionality much greater than 1 (and thus closer to the representational dimension $d \gg 1$). We could regularize by maximizing the sum of squared asymptotic FR distances ($L_2$-style regularization), or of absolute FR distances ($L_1$-style regularization).

In accordance to Corollary 2.1 in Section 4.2, we apply $L_1$-regularization to minimize the negative log of the geometric mean of the ID values. Assuming that $\mathrm{LID}_{F_w^i}^*$ is desired to be $\geq 1$,

$$\max \frac{1}{N} \sum_i^N \lim_{w \to 0} d_{\mathrm{AFR}}(F_w^i(r), U_{1,w}(r)) \ = \ \min -\frac{1}{N} \sum_i^N \ln \mathrm{LID}_{F_w^i}^* \ , \tag{1}$$

where $N$ is the batch size. Following Theorem 2 in Section 4.2, we apply $L_2$-regularization to maximize the Fréchet variance under a prior of $\mu_{\mathcal{F}} = 1$:

$$\max \frac{1}{N} \sum_i^N \lim_{w \to 0} (d_{\mathrm{AFR}}(F_w^i(r), U_{1,w}(r)))^2 \ = \ \min -\frac{1}{N} \sum_i^N \left( \ln \mathrm{LID}_{F_w^i}^* \right)^2 . \tag{2}$$

Our preference for the geometric mean over the arithmetic mean for $L_1$- and $L_2$-regularization is justified by Theorem 4 in Appendix C.5. We refer readers to Appendix D for a discussion of other regularization formulations.

We use the Method of Moments (Amsaleg et al., 2018) as our estimator of LID, due to its simplicity. Since only the encoder is kept for downstream tasks, we estimate the LID values based on the encoder representations ($\mathbf{z} = f(\mathbf{x})$). Specifically, we calculate the pairwise Euclidean distance between the encoder representations of a batch of samples to estimate the $\mathrm{LID}_{F_w^i}^*$ for each sample $\mathbf{x}_i$ in the batch: $\mathrm{LID}_{F_w^i}^* = -\frac{\mu_k}{\mu_k - w_k}$, where $k$ denotes the number of nearest neighbors of $\mathbf{z}_i$, $w_k$ is the distance to the $k$-th nearest neighbor, and $\mu_k$ is the average distance to all $k$ nearest neighbors.

The overall optimization objective is defined as a minimization of either of the following losses:

$$\mathcal{L}_{L1} = \mathcal{L}^{\mathrm{SSL}} - \beta \frac{1}{N} \sum_i^N \ln \mathrm{LID}_{F_w^i}^* \quad \text{or} \quad \mathcal{L}_{L2} = \mathcal{L}^{\mathrm{SSL}} - \beta \left( \frac{1}{N} \sum_i^N \left( \ln \mathrm{LID}_{F_w^i}^* \right)^2 \right)^{\frac{1}{2}} , \tag{3}$$

where $\beta$ is a hyperparameter balancing the loss and regularization terms. More details of how to apply LDReg on different SSL methods can be found in Appendix E; the pseudocode is in Appendix J.

## 6 EXPERIMENTS

We evaluate the performance of LDReg in terms of representation quality, such as training a linear classifier on top of frozen representations. We use SimCLR (Chen et al., 2020a), SimCLR-Tuned (Garrido et al., 2023b), BYOL (Grill et al., 2020), and MAE (He et al., 2022) as baselines. We perform our evaluation with ResNet-50 (He et al., 2016) (for SimCLR, SimCLR-Tuned, and BYOL) and ViT-B (Dosovitskiy et al., 2021) (for SimCLR and MAE) on ImageNet (Deng et al., 2009). As a default, we use batch size 2048, 100 epochs of pretraining for SimCLR, SimCLR-Tuned and BYOL, and 200 epochs for MAE, and hyperparameters chosen in accordance with each baseline's recommended values. We evaluate transfer learning performance by performing linear evaluations on other datasets, including Food-101 (Bossard et al., 2014), CIFAR (Krizhevsky & Hinton, 2009), Birdsnap (Berg et al., 2014), Stanford Cars (Krause et al., 2013), and DTD (Cimpoi et al., 2014). For finetuning, we use RCNN (Girshick et al., 2014) to evaluate on downstream tasks using the COCO dataset (Lin et al., 2014). Detailed experimental setups are provided in Appendix F. For LDReg regularization, we use $k = 64$ as the default neighborhood size. For ResNet-50, we set $\beta = 0.01$ for SimCLR and SimCLR Tuned, $\beta = 0.005$ for BYOL. For ViT-B, we set $\beta = 0.001$ for SimCLR, and $\beta = 5 \times 10^{-6}$ for MAE. Since $\mathcal{L}_{L1}$ and $\mathcal{L}_{L2}$ perform similarly (see Appendix G.1), here we mainly report the results of $\mathcal{L}_{L1}$. An experiment showing how local collapse triggers mode collapse is provided in Appendix G.2, while an ablation study of hyperparameters is in Appendix G.3.

## 6.1 LDReg Regularization Increases Local and Global Intrinsic Dimensions

Intrinsic dimensionality has previously been used to understand deep neural networks in a supervised learning context (Ma et al., 2018b; Gong et al., 2019; Ansuini et al., 2019). For SSL, Figure 2a shows that the geometric mean of LID tends to increase over the course of training. For contrastive methods (SimCLR and BYOL), the mean of LID slightly decreases at the later training stages (dash lines). With LDReg, the mean of LID increases for all baseline methods and alleviates the decreasing trend at the later stages (solid lines), most notably on BYOL.

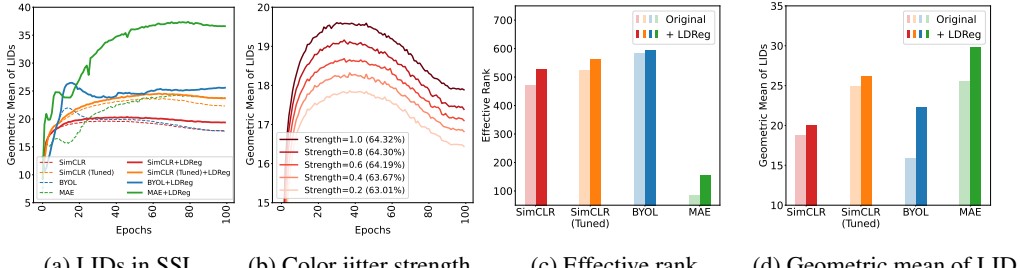

(a) LIDs in SSL      (b) Color jitter strength      (c) Effective rank      (d) Geometric mean of LID

Figure 2: (a) Geometric mean of LID values over training epochs. (b) Geometric mean of LID values with varying color jitter strength in the augmentations for SimCLR. The linear evaluation result is reported in the legend. (a-b) LID is computed on the training set. (c-d) The effective rank and LID are computed for samples in the validation set. The solid and transparent bars represent the baseline method with and without LDReg regularization, respectively. MAE uses ViT-B as the encoder, and others use ResNet-50.

In Figure 2b, we adjusted the color jitter strength of the SimCLR augmentation policy and observed that the mean LID of the representation space positively correlates with the strength. This indicates that stronger augmentations tend to trigger more variations of the image and thus lead to representations of higher LID. This provides insights on why data augmentation is important for SSL Grill et al. (2020); Von Kügelgen et al. (2021) and why it can help avoid dimensional collapse (Wagner et al., 2022; Huang et al., 2023).

The effective rank (Roy & Vetterli, 2007) is a metric to evaluate dimensionality as a global property and can also be used as a metric for representation quality (Garrido et al., 2023a). Figure 2c shows that BYOL is less susceptible to dimensional collapse. SimCLR-Tuned uses the same augmentation as BYOL (which is stronger than that of SimCLR), yet still converges to a lower dimensional space as compared with BYOL. This indicates that for SimCLR and its variants, stronger augmentation is not sufficient to prevent global dimensional collapse. Generative method MAE is known to be prone to dimensional collapse (Zhang et al., 2022). Unsurprisingly, it has the lowest effective rank in Figure 2c. Note that the extremely low effective rank of MAE is also related to its low representation dimension which is 768 in ViT-B (other methods shown here used ResNet-50 which has a representation dimension of 2048).

We also analyze the geometric mean of the LID values in Figure 2d. It shows that, compared to other methods, BYOL has a much lower mean LID value. This implies that although BYOL does not collapse globally, it converges to a much lower dimension locally. We refer readers to Appendix G.2 for an analysis of how local collapse (extremely low LID) could trigger a complete mode collapse, thereby degrading the representation quality. Finally, the use of our proposed LDReg regularization can effectively void dimensional collapse and produce both increased global and local dimensionalities (as shown in Figures 2c and 2d with '+ LDReg').

## 6.2 Evaluations

We evaluate the representation quality learned by different methods via linear evaluation, transfer learning, and fine-tuning on downstream tasks. As shown in Table 1, LDReg consistently improves the linear evaluation performance for methods that are known to be susceptible to dimensional collapse, including sample-contrastive method SimCLR and generative method MAE. It also improves BYOL, which is susceptible to local dimensional collapse as shown in Figure 2d. Tables 2 and 3 further demonstrate that LDReg can also improve the performance of transfer learning, finetuning

Table 1: The linear evaluation results (accuracy (%)) of different methods with and without LDReg. The effective rank is calculated on the ImageNet validation set. The best results are **boldfaced**.

| Model | Epochs | Method | Regularization | Linear Evaluation | Effective Rank | Geometric mean of LID |
|---|---|---|---|---|---|---|
| ResNet-50 | 100 | SimCLR | - | 64.3 | 470.2 | 18.8 |
| | | | LDReg | **64.8** | 529.6 | 20.0 |
| | | SimCLR (Tuned) | - | 67.2 | 525.8 | 24.9 |
| | | | LDReg | **67.5** | 561.7 | 26.1 |
| | | BYOL | - | 67.6 | 583.8 | 15.9 |
| | | | LDReg | **68.5** | 594.0 | 22.3 |
| ViT-B | 200 | SimCLR | - | 72.9 | 283.7 | 13.3 |
| | | | LDReg | **73.0** | 326.1 | 13.7 |
| | | MAE | - | 57.0 | 86.4 | 25.8 |
| | | | LDReg | **57.6** | 154.1 | 29.8 |

Table 2: The transfer learning results in terms of linear probing accuracy (%), using ResNet-50 as the encoder. The best results are **boldfaced**.

| Method | Regularization | Batch Size | Epochs | ImageNet | Food-101 | CIFAR-10 | CIFAR-100 | Birdsnap | Cars | DTD |
|---|---|---|---|---|---|---|---|---|---|---|
| SimCLR | - | 2048 | 100 | 64.3 | 69.0 | 89.1 | **71.2** | 32.0 | 36.7 | **67.8** |
| | LDReg | | | **64.8** | **69.1** | **89.2** | 70.6 | **33.4** | **37.3** | 67.7 |
| | - | 4096 | 1000 | 69.0 | 71.1 | 90.1 | 71.6 | 37.5 | 35.3 | 70.7 |
| | LDReg | | | **69.8** | **73.3** | **91.8** | **75.1** | **38.7** | **41.6** | **70.8** |

Table 3: The performance of the pre-trained models (ResNet-50) on object detection and instance segmentation tasks, when fine-tuned on COCO. The bounding-box ($AP^{bb}$) and mask ($AP^{mk}$) average precision are reported with the best results are **boldfaced**.

| Method | Regularization | Epochs | Batch Size | Object Detection | | | Segmentation | | |
|---|---|---|---|---|---|---|---|---|---|
| | | | | $AP^{bb}$ | $AP^{bb}_{50}$ | $AP^{bb}_{75}$ | $AP^{mk}$ | $AP^{mk}_{50}$ | $AP^{mk}_{75}$ |
| SimCLR | - | 100 | 2048 | 35.24 | 55.05 | **37.88** | 31.30 | 51.70 | 32.82 |
| | LDReg | | | **35.26** | **55.10** | 37.78 | **31.38** | **51.88** | **32.90** |
| BYOL | - | | | 36.30 | 55.64 | 38.82 | 32.17 | 52.53 | 34.30 |
| | LDReg | | | **36.82** | **56.47** | **39.62** | **32.47** | **53.15** | **34.60** |
| SimCLR | - | 1000 | 4096 | 36.48 | 56.22 | 39.28 | 32.12 | 52.70 | 34.02 |
| | LDReg | | | **37.15** | **57.20** | **39.82** | **32.82** | **53.81** | **34.74** |

on object detection and segmentation datasets. Moreover, longer pretraining with LDReg can bring more significant performance improvement. These results indicate that using LDReg to regularize local dimensionality can consistently improve the representation quality.

Table 1 also indicates that the effective rank is a good indicator of representation quality for the same type of SSL methods and the same model architecture. However, the correlation becomes less consistent when compared across different methods. For example, SimCLR + LDReg and SimCLR-Tuned have similar effective ranks ($\sim 525$), yet perform quite differently on ImageNet (with an accuracy difference of 2.4%). Nevertheless, applying our LDReg regularization can improve both types of SSL methods.

# 7 CONCLUSION

In this paper, we have highlighted that dimensional collapse in self-supervised learning (SSL) could occur locally, in the vicinity of any training point. Based on a novel derivation of an asymptotic variant of the Fisher-Rao metric, we presented a local dimensionality regularization method *LDReg* to alleviate dimensional collapse from both global and local perspectives. Our theoretical analysis implies that reporting and averaging intrinsic dimensionality (ID) should be done at a logarithmic (rather than linear) scale, using the geometric mean (but not the arithmetic or harmonic mean). Following these theoretical insights, LDReg regularizes the representation space of SSL to have nonuniform local nearest-neighbor distance distributions, maximizing the logarithm of the geometric mean of the sample-wise LIDs. We empirically demonstrated the effectiveness of LDReg in improving the representation quality and final performance of SSL. We believe LDReg can potentially be applied as a generic regularization technique to help other SSL methods.

## REPRODUCIBILITY STATEMENT

Details of all hyperparameters and experimental settings are given in Appendix F. Pseudocode for LDReg and LID estimation can be found in Appendix J. A summary of the implementation is available in Appendix I. We provide source code for reproducing the experiments in this paper, which can be accessed here: https://github.com/HanxunH/LDReg. We also discuss computational limitations of LID estimation in Appendix H.

## ACKNOWLEDGMENTS

Xingjun Ma is in part supported by the National Key R&D Program of China (Grant No. 2021ZD0112804) and the Science and Technology Commission of Shanghai Municipality (Grant No. 22511106102). Sarah Erfani is in part supported by Australian Research Council (ARC) Discovery Early Career Researcher Award (DECRA) DE220100680. This research was supported by The University of Melbourne's Research Computing Services and the Petascale Campus Initiative.

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

# A ACHIEVING DESIRED LOCAL DIMENSIONALITY WITH LDREG

In this section, we provide details regarding how Figure 1c is obtained. Following our theory, LDReg can obtain representations that have a desired local dimensionality. We use a linear layer and randomly generated synthetic data points in 2D following the uniform distribution. The linear layer transforms these points into a representation space. Following Definition 3, one might specify the desired local intrinsic dimensions as $\text{LID}_G^*$. The dimension of the representations is $\text{LID}_F^*$. To achieve the desired local dimensionality, we minimize the following objective:

$$\min \left( \frac{1}{N} \sum_i^N \ln \frac{\text{LID}_{F_w^i}^*}{\text{LID}_G^*} \right).$$

This objective corresponds to minimizing the asymptotic Fisher-Rao distance between the distribution of the representations and a target distribution of fixed local dimensionality. In Figure 3, we plotted the results with target dimensions $\text{LID}_G^*$ equal to [1.0, 1.2, 1.4, 1.6, 1.8, 2.0]. The results show that the estimated LID is very close to the desired values. This indicates that LDReg can also be used to regularize the representations to a specific value. From this point of view, LDReg can potentially be applied to other learning tasks as a generic representation regularization technique.

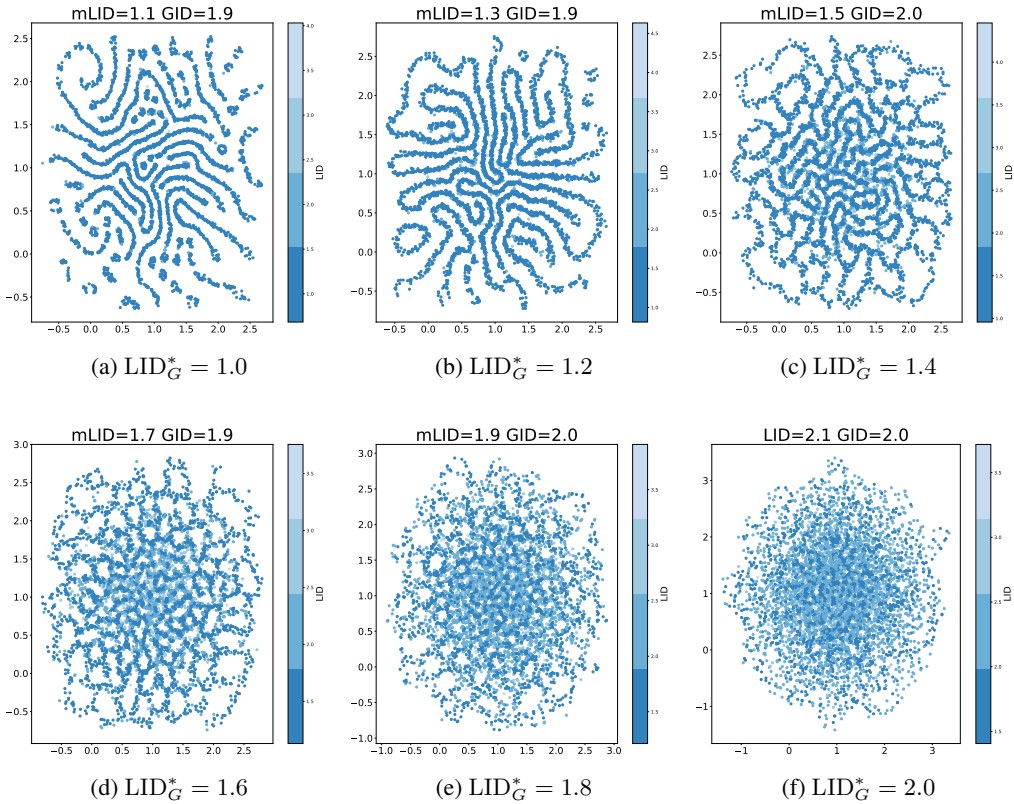

(a) $\text{LID}_G^* = 1.0$  (b) $\text{LID}_G^* = 1.2$  (c) $\text{LID}_G^* = 1.4$

(d) $\text{LID}_G^* = 1.6$  (e) $\text{LID}_G^* = 1.8$  (f) $\text{LID}_G^* = 2.0$

Figure 3: Each caption of the subfigures shows the desired local dimensionality and each title of the subfigures shows the estimated LID and global intrinsic dimensionality (GID). GID is estimated using the DanCO approach (Ceruti et al., 2014). mLID is the geometric mean of estimated sample LIDs.

In the context of SSL, to avoid dimensional collapse, the resulting representation should span (fill) the entire space, as shown in Figure 3f. The $L_1$- and $L_2$-regularization terms used for SSL aim to maximize the asymptotic Fisher-Rao distance between local distance distribution $F(r)$ and a uniform distance distribution $\mathcal{U}_{1,w}(r)$ (which has LID equal to 1, as shown in Figure 3a). In other words, the final representations should be 'further' from that of Figure 3a, and 'closer' to that of Figure 3f.

# B   EFFECTIVE RANK VERSUS LOCAL INTRINSIC DIMENSIONALITY

Zhuo et al. (2023) proposed the effective rank (Roy & Vetterli, 2007) as a metric to evaluate the degree of (global) dimensional collapse. Given the feature correlation matrix, the effective rank corresponds to the exponential of the entropy of the normalized eigenvalues. It is invariant to scaling and takes real values, not just integers. It has a maximum value equal to the representation dimension. Intuitively, the effective rank assesses the degree to which the data 'fills' the representation space, in terms of covariance properties. One might also define a local effective rank, which would assess the degree to which the representation space surrounding a particular anchor sample is filled.

In contrast to the effective rank, the local intrinsic dimension (LID) is a local measure. Roughly speaking, at a particular anchor sample, the LID assesses the growth rate of the distance distribution of nearest neighbors. It is thus focused on distance properties (distances between samples), rather than covariance properties between features. Comparisons between intrinsic dimensionality and effective rank, emphasizing their differences, have been discussed in Del Giudice (2021).

# C   PROOFS

## C.1   BACKGROUND

**Theorem 3** (LID Representation Theorem (Houle, 2017a))**.** *Let $F : \mathbb{R} \to \mathbb{R}$ be a real-valued function, and assume that $\mathrm{LID}_F^*$ exists. Let $r$ and $w$ be values for which $r/w$ and $F(r)/F(w)$ are both positive. If $F$ is non-zero and continuously differentiable everywhere in the interval $[\min\{r, w\}, \max\{r, w\}]$, then*

$$\frac{F(r)}{F(w)} = \left(\frac{r}{w}\right)^{\mathrm{LID}_F^*} \cdot A_F(r, w), \quad where \quad A_F(r, w) \triangleq \exp\left(\int_r^w \frac{\mathrm{LID}_F^* - \mathrm{LID}_F(t)}{t} \, \mathrm{d}t\right),$$

*whenever the integral exists.*

**Corollary 3.1** ((Houle, 2017a))**.** *Let $c$ and $w$ be real constants such that $c \geq 1$ and $w > 0$. Let $F : \mathbb{R} \to \mathbb{R}$ be a real-valued function satisfying the conditions of Theorem 3 over the interval $(0, cw]$. Then*

$$\lim_{\substack{w \to 0^+ \\ w/c \leq r \leq cw}} A_F(r, w) = 1.$$

If $F(0) = 0$, and $F$ is non-decreasing and continuously differentiable over some interval $[0, w]$ for $w > 0$, then $F$ is referred to as a *smooth growth function*. If $F$ is a CDF, we use the notation $F_w$ to refer to $F$ conditioned over $[0, w]$; that is, $F_w(r) = F(r)/F(w)$.

## C.2   PROOF OF LEMMA 1

*Proof.* We leverage a result from (Taylor, 2019), which shows that the Fisher-Rao metric between two one-dimensional distributions with a single parameter can be expressed as $d(\theta_1, \theta_2) = |\int_{\theta_1}^{\theta_2} u(\theta) \, \mathrm{d}\theta|$, where $u(\theta)^2 = \mathcal{I}(\theta)$, and $\mathcal{I}(\theta)$ is the Fisher information with respect to the single parameter $\theta$. In our context, $\theta$ corresponds to the local intrinsic dimensionality. For the Fisher information for functions restricted to the form $H_{w|\theta}$, we will therefore use the quantity

$$\mathcal{I}_w(\theta) = \int_0^w \left(\frac{\partial}{\partial \theta} \ln H'_{w|\theta}(r)\right)^2 H'_{w|\theta}(r) \, \mathrm{d}r.$$

We now derive an expression for the Fisher-Rao distance. From (Taylor, 2019), and noting that Fisher information is greater than or equal to zero,

$$
\begin{aligned}
d_{\mathrm{FR}}(H_{w|\theta_1}, H_{w|\theta_2}) &= \left| \int_{\theta_1}^{\theta_2} \sqrt{\mathcal{I}_w(\theta)} \, \mathrm{d}\theta \right| \\
&= \left| \int_{\theta_1}^{\theta_2} \left( \int_0^w \left( \frac{\partial}{\partial \theta} \ln \left( \frac{\theta}{w} \left( \frac{r}{w} \right)^{\theta-1} \right) \right)^2 \frac{\theta}{w} \left( \frac{r}{w} \right)^{\theta-1} \mathrm{d}r \right)^{\frac{1}{2}} \mathrm{d}\theta \right| \\
&= \left| \int_{\theta_1}^{\theta_2} \left( \int_0^w \left( \frac{1}{\theta} + \ln \frac{r}{w} \right)^2 \frac{\theta}{w} \left( \frac{r}{w} \right)^{\theta-1} \mathrm{d}r \right)^{\frac{1}{2}} \mathrm{d}\theta \right| .
\end{aligned}
$$

With the substitution $v = \left( \frac{r}{w} \right)^{\theta}$, we obtain

$$
\begin{aligned}
d_{\mathrm{FR}}(H_{w|\theta_1}, H_{w|\theta_2}) &= \left| \int_{\theta_1}^{\theta_2} \left( \int_0^1 \left( \frac{1}{\theta} + \frac{1}{\theta} \ln v \right)^2 \mathrm{d}v \right)^{\frac{1}{2}} \mathrm{d}\theta \right| \\
&= \left| \int_{\theta_1}^{\theta_2} \frac{1}{\theta} \sqrt{\left[ v + 2(v \ln v - v) + (v \ln^2 v - 2v \ln v + 2v) \right] \Big|_0^1} \, \mathrm{d}\theta \right| \\
&= \left| \int_{\theta_1}^{\theta_2} \frac{1}{\theta} \, \mathrm{d}\theta \right| = \left| \ln \frac{\theta_2}{\theta_1} \right| .
\end{aligned}
$$

$\square$

### C.3 PROOF OF THEOREM 2

*Proof.* We find the distribution $H_{w|\theta}$ that minimizes the expression

$$
\theta_{\mathrm{G}} = \arg \min_{\theta} \frac{1}{N} \sum_{i=1}^N \left( d_{\mathrm{AFR}}(H_{w|\theta}, F^i) \right)^2 ,
$$

by taking the partial derivative with respect to $\theta$, and solving for the value $\theta = \theta_{\mathrm{G}}$ for which the partial derivative is zero.

$$
\begin{aligned}
\frac{\partial}{\partial \theta} \left( \frac{1}{N} \sum_{i=1}^N \left| \ln \frac{\theta}{\mathrm{LID}_{F^i}^*} \right|^2 \right) \Bigg|_{\theta=\theta_{\mathrm{G}}} &= 0 \\
\frac{\partial}{\partial \theta} \sum_{i=1}^N \left( \ln^2 \theta + \ln^2 \mathrm{LID}_{F^i}^* - 2 \ln \theta \ln \mathrm{LID}_{F^i}^* \right) \Bigg|_{\theta=\theta_{\mathrm{G}}} &= 0 \\
\sum_{i=1}^N \left( \frac{2 \ln \theta_{\mathrm{G}}}{\theta_{\mathrm{G}}} - 2 \frac{\ln \mathrm{LID}_{F^i}^*}{\theta_{\mathrm{G}}} \right) &= 0 \\
N \ln \theta_{\mathrm{G}} &= \sum_{i=1}^N \ln \mathrm{LID}_{F^i}^* \\
\theta_{\mathrm{G}} &= \exp \left( \frac{1}{N} \sum_{i=1}^N \ln \mathrm{LID}_{F^i}^* \right) .
\end{aligned}
$$

In a similar fashion, the second partial derivative can be shown to be strictly positive at $\theta = \theta_1$. Since the original expression is continuous and non-negative over all $\theta \in [0, \infty)$, $\theta_1$ is a global minimum. $\square$

## C.4 PROOF OF COLLARY 2.1

*Proof.* Assertion 1 follows from the fact that

$$
\begin{aligned}
\frac{1}{N} \sum_{i=1}^{N} d_{\mathrm{AFR}}(F^i, \mathcal{U}_1) &= \frac{1}{N} \sum_{i=1}^{N} \left| \ln \frac{\mathrm{LID}^*_{F^i}}{1} \right| \\
&= \frac{1}{N} \sum_{i=1}^{N} \ln \mathrm{LID}^*_{F^i} \quad (\text{if } \mathrm{LID}^*_{F^i} \geq 1) \,.
\end{aligned}
$$

$\square$

## C.5 KULLBACK-LEIBLER DIVERGENCE AS DISTRIBUTIONAL DIVERGENCE

It is natural to consider whether other measures of distributional divergence could be used in place of the asymptotic Fisher-Rao metric, in the derivation of the Fréchet mean. Bailey et al. (2022) have shown several other divergences and distances which, when conditioned to a vanishing lower tail, tend to expressions involving the local intrinsic dimensionalities of distance distributions — most notably that of the Kullback-Leibler (KL) divergence. Here, we define an asymptotic distributional distance from the square root of the asymptotic KL divergence considered in (Bailey et al., 2022).

**Lemma 2** ((Bailey et al., 2022)). *Given two smooth-growth distance distributions with CDFs F and G, their* asymptotic KL distance *is given by*

$$
d_{\mathrm{AKL}}(F, G) \triangleq \lim_{w \to 0^+} \sqrt{D_{\mathrm{KL}}(F_w, G_w)} = \sqrt{\frac{\mathrm{LID}^*_G}{\mathrm{LID}^*_F} - \ln \frac{\mathrm{LID}^*_G}{\mathrm{LID}^*_F} - 1} \,,
$$

*where*

$$
D_{\mathrm{KL}}(F_w, G_w) \triangleq \int_0^w F'_w(t) \ln \frac{F'_w(t)}{G'_w(t)} \, \mathrm{d}t
$$

*is the KL divergence from F to G when conditioned to the lower tail $[0, w]$.*

When aggregating the LID values of distance distributions, it is worth considering how well the arithmetic mean (equivalent to the information dimension (Romano et al., 2016)) and the harmonic mean might serve as alternatives to the geometric mean. Theorem 4 shows that the arithmetic and harmonic means of distributional LIDs are obtained when the asymptotic Fisher-Rao metric is replaced by the asymptotic KL distance, in the derivation of the Fréchet mean.

**Theorem 4.** *Given a set of distributions $\mathcal{F} = \{F^1, F^2, \ldots, F^N\}$, consider the metric used in computing the Fréchet mean $\mu_{\mathcal{F}} = H_{w|\theta}$.*

1. *Using the asymptotic Fisher-Rao metric $d_{\mathrm{AFR}}(H_{w|\theta}, F^i)$ as in Definition 4 gives $\theta$ equal to the geometric mean of $\{\mathrm{LID}^*_{F^1_w}, \ldots, \mathrm{LID}^*_{F^N_w}\}$.*

2. *Replacing $d_{\mathrm{AFR}}(H_{w|\theta}, F^i)$ by the asymptotic KL distance $d_{\mathrm{AKL}}(H_{w|\theta}, F^i)$ gives $\theta$ equal to the arithmetic mean of $\{\mathrm{LID}^*_{F^1}, \ldots, \mathrm{LID}^*_{F^N}\}$.*

3. *Replacing $d_{\mathrm{AFR}}(H_{w|\theta}, F^i)$ by the (reverse) asymptotic KL distance $d_{\mathrm{AKL}}(F^i, H_{w|\theta})$ gives $\theta$ equal to the harmonic mean of $\{\mathrm{LID}^*_{F^1}, \ldots, \mathrm{LID}^*_{F^N}\}$.*

*Proof.* Assertion 1 has been shown in Theorem 2.

For Assertion 2, we find the value of $\theta$ for which the following expression is minimized:

$$
\theta_{\mathrm{A}} = \arg\min_{\theta} \frac{1}{N} \sum_{i}^{N} \left( d_{\mathrm{AKL}}(H_{w|\theta}, F^i) \right)^2 \,.
$$

Using Lemma 2, and observing that $\mathrm{LID}^*_{H_{w|\theta}} = \theta$,

$$
\left( d_{\mathrm{AKL}}(H_{w|\theta}, F^i) \right)^2 = \lim_{w \to 0} D_{\mathrm{KL}}(H_{w|\theta}, F^i) = \frac{\mathrm{LID}^*_{F^i}}{\theta} - \ln \frac{\mathrm{LID}^*_{F^i}}{\theta} - 1 \,.
$$

As in the proof of Theorem 2, the minimization is accomplished by setting the partial derivative to zero and solving for $\theta = \theta_A$.

$$\frac{\partial}{\partial \theta} \left( \frac{1}{N} \sum_{i=1}^{N} \left( \frac{\text{LID}_{F^i}^*}{\theta} - \ln \frac{\text{LID}_{F^i}^*}{\theta} - 1 \right) \right) \Bigg|_{\theta = \theta_A} = 0$$

$$\frac{1}{N} \sum_{i=1}^{N} \left( -\frac{\text{LID}_{F^i}^*}{\theta_A^2} + \frac{1}{\theta_A} \right) = 0$$

$$\theta_A = \frac{1}{N} \sum_{i=1}^{N} \text{LID}_{F^i}^* .$$

For Assertion 3, we similarly find the value of $\theta$ for which the following expression is minimized:

$$\theta_H = \arg\min_{\theta} \frac{1}{N} \sum_{i}^{N} \left( d_{\text{AKL}}(F^i, H_{w|\theta}) \right)^2 .$$

Once again, we take the partial derivative with respect to $\theta$, set it to zero, and solve for $\theta = \theta_H$:

$$\frac{\partial}{\partial \theta} \left( \frac{1}{N} \sum_{i=1}^{N} \left( \frac{\theta}{\text{LID}_{F^i}^*} - \ln \frac{\theta}{\text{LID}_{F^i}^*} - 1 \right) \right) \Bigg|_{\theta = \theta_H} = 0$$

$$\frac{1}{N} \sum_{i=1}^{N} \left( \frac{1}{\text{LID}_{F^i}^*} - \frac{1}{\theta_H} \right) = 0$$

$$\theta_H = \frac{N}{\sum_{i=1}^{N} \frac{1}{\text{LID}_{F^i}^*}} .$$

Note that $\theta_G$ and $\theta_H$ can be verified as minima by computing the second partial derivatives with respect to $\theta$. $\qquad\square$

The square root of the KL divergence is only a 'weak approximation' of the Fisher-Rao metric on statistical manifolds, and this approximation is known to degrade as distributions diverge (Carter et al., 2007). Moreover, the KL divergence is also nonsymmetric. Theorem 4 therefore indicates that the asymptotic Fisher-Rao metric is preferable (in theory) to the asymptotic KL distance, and the geometric mean is preferable to the arithmetic mean and harmonic mean when aggregating the LID values of distance distributions.

## D    OTHER REGULARIZATION FORMULATIONS

We elaborate on our choices of regularization term.

**Remark 4.1.** *The proposed regularization is equivalent to maximizing the (log of the) geometric mean of the IDs of the samples.*

Theorem 4 provided arguments for why use of geometric mean is preferable to other means for the purpose of computing the Frechet mean of a set of distributions. We can similarly consider why a regularization corresponding to the arithmetic mean of the LIDs of the samples would be less preferable. i.e. $\max \frac{1}{N} \sum_{i}^{N} \text{LID}_{F_w^i}^*$. Note that

$$\max \frac{1}{N} \sum_{i}^{N} \text{LID}_{F_w^i}^* = \max \frac{1}{N} \sum_{i}^{N} \exp \left( \lim_{w \to 0} d_{FR}(F_w^i(r), U_{1,w}(r)) \right) .$$

Observe that this arithmetic mean regularization, due to the exponential transformation, would apply a high weighting to samples with very large distances from the uniform distribution (that is, samples with large ID). In other words, such a regularization objective could be optimized by making the ID of a small number of samples extremely large.

# E  LDReg and SSL methods

In this section, we provide more details on how to apply LDReg on different SSL methods. Since LDReg is applied to the representation obtained by the encoder, the varying combinations of projector, predictor, decoder, and optimizing objective used by SSL methods does not directly affect how LDReg is applied. As a result, LDReg can be regarded as a general regularization for SSL.

**SimCLR.** For input images $\mathbf{x}$ of batch size $N$, an encoder $f(\cdot)$, and a projector $g(\cdot)$, the representations are obtained by $\boldsymbol{z} = f(\mathbf{x})$, and the embeddings are $\boldsymbol{e} = g(\boldsymbol{z})$. Given a batch of $2N$ augmented inputs, the NT-Xent loss used by SimCLR (Chen et al., 2020a) for a positive pair of inputs $(\mathbf{x}_i, \mathbf{x}_j)$ is:

$$\mathcal{L}_i^{\text{NTXent}} = -\ln \frac{\exp(\text{sim}(\boldsymbol{e_i}, \boldsymbol{e_j})/\tau)}{\sum_{m \neq j}^{2N} \exp(\text{sim}(\boldsymbol{e_i}, \boldsymbol{e_m})/\tau)},$$

where $\tau$ is the temperature, and the final loss is computed across all positive pairs.

For applying LDReg with $\mathcal{L}_{L1}$ term on SimCLR, we optimize the following objective:

$$\mathcal{L}_{L1} = \mathcal{L}^{\text{NTXent}} - \beta \frac{1}{2N} \sum_i^{2N} \ln \text{LID}_{F_w^i}^*,$$

where the LID for each sample is estimated using the method of moments: $\text{LID}_{F_w^i}^* = -\frac{\mu_k}{\mu_k - w_k}$, where $\mu_k$ is the averaged distance to the $k$ nearest neighbors of $\boldsymbol{z}_i$, and $w_k$ is the distance to the $k$-th nearest neighbor of $\boldsymbol{z}_i$.

**BYOL.** BYOL (Grill et al., 2020) uses an additional predictor $h(\cdot)$ to obtain predictions $\boldsymbol{p} = h(\boldsymbol{e})$, a momentum encoder (exponential moving average of the weights of the online model) where $\boldsymbol{e}'$ is obtained, and the loss function is the scaled cosine similarity between positive pairs, defined as:

$$\mathcal{L}_i^{\text{BYOL}} = 2 - 2 \frac{\boldsymbol{p_i} \cdot \boldsymbol{e_j'}}{\|\boldsymbol{p_i}\|_2 \|\boldsymbol{e_j'}\|_2},$$

with the final loss computed symmetrically across all positive pairs.

For applying LDReg with $\mathcal{L}_{L1}$ term on BYOL, we optimize the following objective:

$$\mathcal{L}_{L1} = \mathcal{L}^{\text{BYOL}} - \beta \frac{1}{2N} \sum_i^{2N} \ln \text{LID}_{F_w^i}^*.$$

The LID for each sample is estimated in the same way as applying LDReg on SimCLR. For BYOL, we use representations obtained by both the online and momentum branches as the reference set.

**MAE.** MAE (He et al., 2022) uses a decoder that aims to reconstruct the input image. Unlike a contrastive approach, it does not rely on two different augmented views of the same image. MAE uses an encoder $f(\cdot)$ to obtain the representation $\boldsymbol{z} = f(\mathbf{x}')$ for a masked image $\mathbf{x}'$, and the decoder $t(\cdot)$ aims to reconstruct the original image $\mathbf{x}$ by taking representation $\boldsymbol{r}$ as input. Specifically, MAE optimizes the following objective:

$$\mathcal{L}_i^{\text{MAE}} = \|t(f(\mathbf{x}_i')), \mathbf{x}_i\|_2.$$

For applying LDReg with $\mathcal{L}_{L1}$ term on MAE, we optimize the following objective:

$$\mathcal{L}_{L1} = \mathcal{L}^{\text{MAE}} - \beta \frac{1}{N} \sum_i^N \ln \text{LID}_{F_w^i}^*.$$

The LID for each sample is estimated in the same way as applying LDReg on SimCLR.

# F  Experimental Settings

For each baseline method, we follow their original settings, except in the case of BYOL, where we changed the parameter for the exponential moving average from 0.996 to 0.99, which performs

better when the number of epochs is set to 100. Detailed hyperparameter settings can be found in Tables 5-11. We use 100 epochs of pretraining and a batch size of 2048 as defaults. For LDReg regularization, we use $k = 128$ as the default neighborhood size. For ResNet-50, we use $\beta = 0.01$ for SimCLR and SimCLR Tuned, $\beta = 0.005$ for BYOL. For ViT-B, we use $\beta = 0.001$ for SimCLR, and $\beta = 5 \times 10^{-6}$ for MAE. We perform linear evaluations following existing works (Chen et al., 2020a; Grill et al., 2020; He et al., 2022; Garrido et al., 2023b). For linear evaluations, we use batch size 4096 on ImageNet — other settings are shown in Table 9.

Following SimCLR (Chen et al., 2020a) and BYOL (Grill et al., 2020), we evaluate transfer learning performance by performing linear evaluations on other datasets, including Food-101 (Bossard et al., 2014), CIFAR (Krizhevsky & Hinton, 2009), Birdsnap (Berg et al., 2014), Stanford Cars (Krause et al., 2013), and DTD (Cimpoi et al., 2014). Due to computational constraints, for the transfer learning experiments, we did not perform full hyperparameter tuning for each model and dataset. The reproduced results of baseline methods are slightly lower than the reported results by Chen et al. (2020a); Grill et al. (2020). For all datasets, we use 30 epochs, weight decay to 0.0005, learning rate 0.01, batch size 256, and SGD with Nesterov momentum as optimizer. These settings are based on the `VISSL` library [2].

We evaluate the finetuning performance with downstream tasks using the COCO dataset (`train2017` and `val2017`) (Lin et al., 2014). We use ResNet-50 with RCNN-C4 (Girshick et al., 2014) with batch size 16 and base learning rate 0.02. We use the popular framework `detectron2` [3], and our configurations follow the `MOCO-v1` official implementation [4] exactly.

We conducted our experiments on Nvidia A100 GPUs with PyTorch implementation, with each experiment distributed across 4 GPUs. We used automatic mixed precision due to its memory efficiency. The estimated runtime is 40 hours for pretraining and linear evaluations. As can be seen from the pseudocode in Appendix J, the additional computation mainly depends on the calculation and sorting of pairwise distances. As shown in Table 4, we observed no significant additional computational costs for LDReg. Open source code is available here: https://github.com/HanxunH/LDReg.

Table 4: Wall-clock comparisons for pretraining with Distributed Data-Parallel training. Each experiment uses 4 GPUs distributed over different nodes. Results are based on 100 epochs of pretraining. Communication overheads could have a slight effect on the results.

| Method | Wall-clock time |
|---|---|
| SimCLR | 27.8 hours |
| SimCLR + LDReg | 27.1 hours |

Table 5: Pretraining setting for SimCLR  (Chen et al., 2020a).

| | |
|---|---|
| Base learning rate | 0.075 |
| Learning rate scaling | $0.075 \times \sqrt{BatchSize}$ |
| Learning rate decay | Cosine (Loshchilov & Hutter, 2016) without restart |
| Weight Decay | $1.0 \times 10^{-6}$ |
| Optimizer | LARS (You et al., 2017) |
| Temperature for $\mathcal{L}^{\mathrm{NTXent}}$ | 0.1 |
| Projector | 2048-128 |

**Data Augmentations.** For each baseline and LDReg version, we use the same augmentation as in existing works. Augmentation policy for SimCLR (Chen et al., 2020a) is in Table 10, for SimCLR-Tuned (Garrido et al., 2023b) and BYOL (Grill et al., 2020) is in Table 11.

---

[2]https://github.com/facebookresearch/vissl

[3]https://github.com/facebookresearch/detectron2

[4]https://github.com/facebookresearch/moco/tree/main/detection/configs

Table 6: Pretraining setting for SimCLR-Tuned (Garrido et al., 2023b).

| | |
|---|---|
| Base learning rate | 0.5 |
| Learning rate scaling | $0.5 \times \frac{BatchSize}{256}$ |
| Learning rate decay | Cosine (Loshchilov & Hutter, 2016) without restart |
| Weight Decay | $1.0 \times 10^{-6}$ |
| Optimizer | LARS (You et al., 2017) |
| Temperature for $\mathcal{L}^{\mathrm{NTXent}}$ | 0.15 |
| Projector | 8192-8192-512 |

Table 7: Pretraining setting for BYOL (Grill et al., 2020).

| | |
|---|---|
| Base learning rate | 0.4 |
| Learning rate scaling | $0.4 \times \frac{BatchSize}{256}$ |
| Learning rate decay | Cosine (Loshchilov & Hutter, 2016) without restart |
| Weight Decay | $1.5 \times 10^{-6}$ |
| Optimizer | LARS (You et al., 2017) |
| $\tau$ for moving average | 0.99 |
| Projector | 4096-256 |
| Predictor | 4096-256 |

Table 8: Pretraining setting for MAE (He et al., 2022).

| | |
|---|---|
| Base learning rate | $1.5 \times 10^{-4}$ |
| Learning rate scaling | $1.5 \times 10^{-4} \times \frac{BatchSize}{256}$ |
| Learning rate decay | Cosine (Loshchilov & Hutter, 2016) without restart |
| Weight Decay | 0.05 |
| Optimizer | AdamW (Loshchilov & Hutter, 2019) |
| $\beta_1$ for the optimizer | 0.9 |
| $\beta_2$ for the optimizer | 0.95 |

Table 9: Linear evaluation setting for ImageNet.

| | |
|---|---|
| Epochs | 90 |
| Base learning rate | 0.1 |
| Learning rate scaling | $0.1 \times \frac{BatchSize}{256}$ |
| Minimal learning rate | $1.0 \times 10^{-6}$ |
| Learning rate decay | Cosine (Loshchilov & Hutter, 2016) without restart |
| Weight Decay | 0 |
| Optimizer | LARS (You et al., 2017) |

Table 10: Image augmentation policy for SimCLR (Chen et al., 2020a).

| Parameter | View 1 | View 2 |
|---|---|---|
| Random crop probability | 1.0 | 1.0 |
| Horizontal flip probability | 0.5 | 0.5 |
| Color jittering probability | 0.8 | 0.8 |
| Color jittering strength ($s$) | 1.0 | 1.0 |
| Brightness adjustment max intensity | $0.8 \times s$ | $0.8 \times s$ |
| Contrast adjustment max intensity | $0.8 \times s$ | $0.8 \times s$ |
| Saturation adjustment max intensity | $0.8 \times s$ | $0.8 \times s$ |
| Hue adjustment max intensity | $0.2 \times s$ | $0.2 \times s$ |
| Grayscale probability | 0.2 | 0.2 |
| Gaussian blurring probability | 0.5 | 0.5 |

Table 11: Image augmentation policy for BYOL (Grill et al., 2020) and SimCLR-Tuned (Garrido et al., 2023b).

| Parameter | View 1 | View 2 |
|---|---|---|
| Random crop probability | 1.0 | 1.0 |
| Horizontal flip probability | 0.5 | 0.5 |
| Color jittering probability | 0.8 | 0.8 |
| Brightness adjustment max intensity | 0.4 | 0.4 |
| Contrast adjustment max intensity | 0.4 | 0.4 |
| Saturation adjustment max intensity | 0.2 | 0.2 |
| Hue adjustment max intensity | 0.1 | 0.1 |
| Grayscale probability | 0.2 | 0.2 |
| Gaussian blurring probability | 1.0 | 0.1 |
| Solarization probability. | 0.0 | 0.2 |

# G  ADDITIONAL EXPERIMENTAL RESULTS

## G.1  COMPARING LOSS TERMS

It can be observed from Table 12 that there are no significant differences between the regularization terms $\mathcal{L}_{L1}$ and $\mathcal{L}_{L2}$ for improving the performance of SSL.

Table 12: Comparing the results of linear evaluations of regularization terms of LDReg. All models are trained on ImageNet for 100 epochs. The results are reported as linear probing accuracy (%).

| Method | Regularization | k=64 | k=128 |
|---|---|---|---|
| SimCLR | $\mathcal{L}_{L1}$ | 64.8 | 64.6 |
| | $\mathcal{L}_{L2}$ | 64.4 | 64.5 |

## G.2  LOCAL COLLAPSE TRIGGERING COMPLETE COLLAPSE

In this section, we demonstrate that local collapse could trigger the worst-case mode collapse, where the output representation is a trivial vector. LDReg is a general regularization tool that regularizes representation to achieve a target LID. One can also use LDReg to achieve lower LID and, in the extreme case, local collapse. Specifically, we optimize the following objective function:

$$\min \left( \frac{1}{N} \sum_{i}^{N} \ln \text{LID}^{*}_{F^i_w} \right).$$

This objective regularizes the geometric mean of LID (of the representations) to 1. We use $\beta$ to control the strength of the regularization term and denote it as *Min LID*.

Table 13: Comparing the results of linear evaluations of regularization terms of LDReg, MinLID and baseline. All models are trained on ImageNet for 100 epochs using ResNet-50 as encoder. The results are reported as linear probing accuracy (%) on ImageNet.

| Method | Regularization | $\beta$ | Linear Acc | Effective Rank | Geometric mean of LID |
|---|---|---|---|---|---|
| SimCLR | LDReg | 0.01 | 64.8 | 529.6 | 20.0 |
| | - | - | 64.3 | 470.2 | 18.8 |
| | Min LID | 0.01 | 64.2 | 150.7 | 16.0 |
| | Min LID | 0.1 | 63.1 | 15.0 | 3.8 |
| | Min LID | 1.0 | 46.4 | 1.0 | 1.6 |
| | Min LID | 10.0 | Complete collapse | - | - |

As shown in Table 13, it can be observed that using *Min LID* with stronger (larger $\beta$) will regularize the representation to have extremely low effective rank and eventually result in complete collapse, even if SimCLR explicitly uses negative pairs to prevent this. Figure 4 shows the visualizations

of learned representations with different regularizations. Additionally, the performance of linear evaluations degrades as dimensionality decreases. This result indicates that low LID is undesirable for SSL.

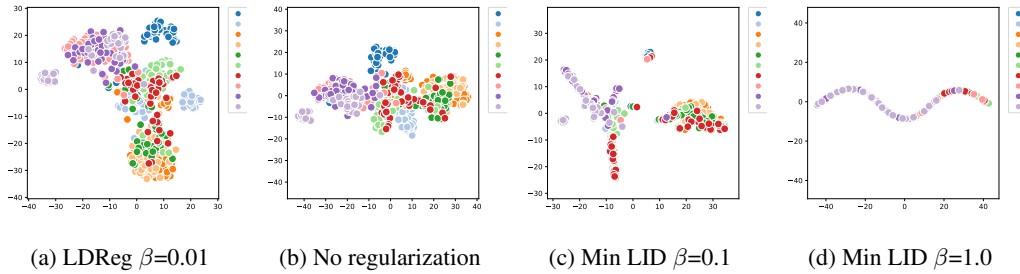

(a) LDReg $\beta$=0.01          (b) No regularization          (c) Min LID $\beta$=0.1          (d) Min LID $\beta$=1.0

Figure 4: t-SNE visualizations of the representations learned by different pretraining. Results are based on ResNet-50 with SimCLR with ImageNet validation set. Only the first 10 classes are selected for visualizations.

### G.3 ABLATION STUDY

We examine the effects of varying $\beta$ and $k$ for LDReg using SimCLR as the baseline. It can be observed in Figure 5a and 5b that linear evaluation performance is relatively stable across different values of $k$ and $\beta$. For effective rank, Figure 5c shows that greater strength of LDReg regularization (larger $\beta$) actually decreases the effective rank. This is not surprising, since LID is a local measure, and effective rank is a global measure. The differences between effective rank and LID are outlined in Appendix B. Figure 5d shows that a smaller value for $k$ is more beneficial for LDReg. Smaller $k$ is indeed more preferable, as it helps to preserve the locality assumptions upon which LID estimation depends.

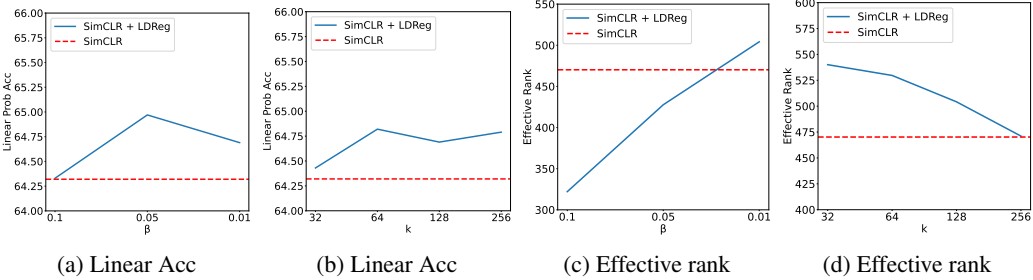

(a) Linear Acc          (b) Linear Acc          (c) Effective rank          (d) Effective rank

Figure 5: (a-b) Linear evaluation results and (c-d) effective ranks with varying $\beta$ and $k$. All models are trained on ImageNet for 100 epochs. The results are reported as linear probing accuracy (%) on ImageNet.

Table 14 shows that LDReg can consistently improve the baseline with different batch sizes. We reduce the $k$ at the same rate as $N$ is reduced, e.g. 16 and 32 for batch sizes 512 and 1024, respectively. It can be observed that LDReg can consistently improve the baseline in different batch sizes.

Table 14: Comparing the results of linear evaluations with different batch sizes $N$. All models are trained on ImageNet for 100 epochs. The results are reported as linear probing accuracy (%).

| Method | Regularization | N=512 | N=1024 | N=2048 |
|---|---|---|---|---|
| SimCLR | - | 63.6 | 64.2 | 64.3 |
|  | LDReg | **64.1** | **64.7** | **64.8** |

## G.4 ADDITIONAL LINEAR EVALUATION RESULTS

In this subsection, we further verify the effectiveness of LDReg with decorrelating feature methods such as VICReg (Bardes et al., 2022) and Barlow Twins (Zbontar et al., 2021). We also evaluate with another SOTA sample-contrastive method MoCo (He et al., 2020). For applying LDReg on VICReg, we use $\beta = 0.025$ and $k = 64$. For Barlow Twins, we use $\beta = 1.0$ and $k = 64$. For MoCo, we use $\beta = 0.05$ and $k = 128$. All other settings are kept the same as each baseline's originally reported hyperparameters. Results can be found in Table 15.

Table 15: The linear evaluation results (accuracy (%)) of different methods with and without LDReg. The effective rank is calculated on the ImageNet validation set. The best results are **boldfaced**.

| Model | Epochs | Method | Regularization | Linear Evaluation | Effective Rank | Geometric mean of LID |
|---|---|---|---|---|---|---|
| ResNet-50 | 100 | MoCo | - | 68.7 | 595.0 | 17.1 |
| | | | LDReg | **69.6** | 651.8 | 22.3 |
| | | VICReg | - | 66.7 | 546.7 | 21.5 |
| | | | LDReg | **66.9** | 602.4 | 22.5 |
| | | Barlow Twins | - | 65.5 | 602.1 | 20.8 |
| | | | LDReg | **65.6** | 754.0 | 24.1 |

VICReg and Barlow Twins are SSL methods rather than regularizers. For example, compared to MoCo, VICReg and Barlow Twins use different projector architectures and loss functions. It's not fair to compare LDReg across different types of SSL methods. For example, MoCo with LDReg has a linear evaluation of 69.6, yet MoCo alone can achieve 68.7, while VICReg under the same setting is only 66.7. However, if we apply our regularizer to these methods, their performance can all be improved, as shown in the table 15.

To fairly compare the regularizers, we use the covariance (denoted as Cov) and variance (denoted as Var) as alternative regularizers to replace LDReg and apply to BYOL. Note that they are pseudo-global dimension regularizers, as we cannot use the entire training set to calculate the covariance, it is calculated on a mini-batch. We also performed a hyperparameter search for Cov and Var ($\beta$ for the strength of the regularization). We used the same regularization formula as VICReg (Bardes et al., 2022) for $Cov$ and $Var$ as the following:

$$C(Z) = \frac{1}{n-1} \sum_{i=1}^{n} (z_i - \bar{z})(z_i - \bar{z})^T, \quad \bar{z} = \frac{1}{n} \sum_{i=1}^{n} z_i, \tag{4}$$

$$Cov(Z) = c(Z) = \frac{1}{d} \sum_{i \neq j} [C(Z)]_{i,j}^2, \tag{5}$$

where $d$ is the representation dimensions.

$$Var(Z) = \frac{1}{d} \sum_{j=1}^{d} \max(0, \gamma - S(z^j, \epsilon)) \quad \text{where } S(\cdot) \text{ is the standard deviation.} \tag{6}$$

We apply the regularization on representations learned by the encoder, the same as in LDReg. The results can be found in the Table 16. All results are based on 100 epoch pretraining with BYOL and ResNet-50. All settings are exactly the same as LDReg except for the regularization term.

Table 16: The linear evaluation results on comparing different regularization terms. The effective rank is calculated on the ImageNet validation set. The best results are **boldfaced**.

| Method | Regularizer | $\beta$ | Linear Evaluation | Effective Rank | Geometric mean of LID |
|---|---|---|---|---|---|
| BYOL | None | - | 67.6 | 583.8 | 15.9 |
| | Cov | 0.01 | 67.6 | 583.5 | 15.9 |
| | Cov | 0.1 | 67.5 | 593.5 | 15.8 |
| | Cov+Var | 0.01 | 67.8 | 539.2 | 15.5 |
| | Cov+Var | 0.1 | 67.7 | 798.4 | 16.8 |
| | LDReg | 0.005 | **68.5** | 594.0 | 22.3 |

Although the covariance and variance regularizer can increase the global dimension, it does not improve the local dimension. It also has a rather minor effect on the linear evaluation accuracy, whereas LDReg improves by almost 1%. This further confirms the effectiveness of LDReg.

## H  LIMITATIONS AND FUTURE WORK

The main theory of our work is based on the local intrinsic dimensionality model and well-founded Fisher-Rao metric. Computation of LDReg and the Fisher-Rao metric all require the ability to accurately estimate the local intrinsic dimension. We have used the method of moments in this paper for LID estimation, due to its simplicity and attractiveness for incorporation within a gradient descent framework. Other estimation methods could be used instead; however, all estimation methods for LID are known to degrade in performance as the dimensionality increases. Moreover, our estimation of LID is based on nearest neighbor sets computed from within a minibatch. This choice is made due to feasibility of computation, but entails a reduction in accuracy as compared to using nearest neighbors computed from the whole dataset. In future work, one might explore other estimation methods and tradeoffs between estimation accuracy and computation time.

Based on existing works, LDReg assumes that higher dimensionality is desirable for SSL. LDReg relies on a hyperparameter $\beta$ to adjust the strength of the regularization term. The theory developed in this work allows LDReg to achieve any desired dimensionality. However, the optimal dimensionalities for SSL are dependent on the dataset and loss function. Knowledge of the optimal dimensionality (if it could be determined) can be integrated into LDReg for best performance.

## I  IMPLEMENTATION DETAILS

For implementation with PyTorch, Garrido et al. (2023b) have discussed popular open-source implementations of SimCLR (compatible with DDP using `gather`) that use slightly inaccurate gradients. The implementation in VICReg (Bardes et al., 2022) codebase [5] is correct and should be used. We find that this slightly affects the performance when reproducing SimCLR's results. This also affects LDReg for estimating LIDs with DDP. For all of our experiments, we use the same implementation as Garrido et al. (2023b); Bardes et al. (2022).

Estimating LID needs to compute the pairwise distance, in PyTorch, the `cdist` function by default uses a matrix multiplication approach. For Nvidia Ampere or newer GPUs, `TensorFloat-32 tensor cores` should be disabled due to precision loss in the matrix multiplication. This precision loss can significantly affect the LID estimations.

## J  PSEUDOCODE

**Algorithm 1:** Method of moments for LID estimation using pytorch pseudocode.

```
# data: representations
# reference: reference points
# k: the number of nearest neighbours

def lid_mom_est(data, reference, k):
    r = torch.cdist(data, reference, p=2) # Pairwise distance
    a, idx = torch.sort(r, dim=1)
    m = torch.mean(a[:, 1:k], dim=1) # mu_k
    lids = m / (a[:, k] - m) # a[:, k] is the w_k
    return lids
```

---

[5]https://github.com/facebookresearch/vicreg

**Algorithm 2:** LDReg using pytorch pseudocode.

```
# f: representations
# k: the number of nearest neighbours
# beta: the hyperparameter $\beta$
# loss: SSL loss (such as NTXent)
# reg_type: "l1" or "l2" (L1 or L2 loss)

lids = lid_mom_est(data=f, reference=f.detach(), k=k)
if reg_type == "l1":
    lid_reg = - torch.abs(torch.log(lids))
elif reg_type == "l2":
    lid_reg = - torch.sqrt(torch.square(torch.log(lids)))
total_loss = loss + beta * lid_reg
total_loss = total_loss.mean(dim=0)
```

