# OpenReview forum: "LDReg: Local Dimensionality Regularized Self-Supervised Learning"
_ICLR.cc/2024/Conference — ICLR 2024 poster_

### Official Review · Reviewer_85fu · 2023-10-26

**Soundness:** 3 good
**Presentation:** 3 good
**Contribution:** 3 good
**Rating:** 6
**Confidence:** 4

**Summary:**

This work introduces a novel approach called LDReg for addressing SSL problems. LDReg incorporates the concept of local intrinsic dimensionality (LID), as measured in (Houle, 2017a), and utilizes it as a penalty term in various SSL models.
The experimental results presented in this paper demonstrate the effectiveness of LDReg in improving SSL performance. The conducted experiments on benchmark datasets highlight the positive impact of incorporating LID as a regularization term.

**Strengths:**

1. The dimensional collapse problem in SSL has received significant attention in recent years, with several works addressing this issue. This work offers a fresh perspective on the problem and introduces a generalized regularization approach that can be applied to existing SSL models.

2. An advantage of this work is its convenient implementation, allowing for improved performance in downstream tasks without the need to modify the underlying architecture. By providing an additional viewpoint on the dimension collapse problem, this work contributes to the ongoing efforts in tackling this challenge.

**Weaknesses:**

1.While the performance of SSL models with LDReg is impressive, the novelty of this work appears to be limited as the proposed LID is inspired by or defined similarly to IntrDim proposed in (Houle, 2017a). Without proper attribution to (Houle, 2017a), the novelty of this work becomes questionable.

2.It is recommended that the authors verify whether Houle et al. have indeed proposed Theorem 1, which defines LID. If not, it would not be reasonable to cite (Houle, 2017a) in Theorem 1 and define LID based on it. On the other hand, if Houle et al. did define LID in (Houle, 2017a), the novelty of this work would be further discounted.

3.Since the only contribution of this work seems to be the definition of LID, which is directly borrowed from (Houle, 2017a), it is suggested to provide a more detailed explanation in Section 3 to clarify the relationship between the proposed LID and the work of (Houle, 2017a).

4.The experimental results lack in-depth analysis. Several significant observations, as mentioned in the Q4, have been overlooked. It is recommended to address these observations and provide a thorough analysis of the experimental results.

**Questions:**

1.This paper appears to have been organized in a hurry, as evidenced by the missing parenthesis in Definition 1. On page 19, there are two versions of $w_k$ ($w^k$).

2.On page 4, it is unclear what ‘Pr’ denotes. Please provide an explanation or definition for this term.

3.In addition to the $\bf{LID}$ defined using $\bf{IntrDim}$ in (Houle, 2017a) and the theoretical analysis using Fisher-Rao metric to provide justification, please clearly highlight the additional contributions of this work.

4.While the experimental results presented on different datasets show improvements compared to other SSL models without LDReg, the authors did not analyze the reasons behind the under-performance results. Additionally, it is unclear how dimensional collapse was observed in the experiment, as the results alone cannot directly reflect this improvement.

5.Technically, the regularization used is $\mu_k / (\mu_k - w_k)$, which prefers that nearest neighbors are not in the same surface of the sphere ball of the sample. Other works that aim to improve dimensional collapse are referred to as compared methods in the experiments.

---

> ### Author Response · Authors · 2023-11-17
> **Response to Reviewer 85fu (1/3)**
>
> Thank you very much for your valuable comments. We hope the following clarifications can help address the raised issues.
>
> ---
> **Q1**: the novelty of this work and its relation to existing work (Houle, 2017a), weaknesses No.1 mentioned by the reviewer.
>
> **A1**: Our work is based on the LID Definition 1, Theorem 1, and Theorem 3 (LID Representation Theorem) by Houle, 2017a. We successfully extended the existing theory to compare and regularize LID distributions. In Section 4, Lemma 1, Corollary 3.1, Definitions 3 and 4, Remark 1.1, and 1.2, Theorem 2 and 4, Corollary 2.1. are all our **new** and **novel** contributions to LID theory. The entire section 4, except for the introduction of the Fisher-Rao metric, is all new theoretical contributions. To the best of our knowledge, the closest related work to ours is the Kullback-Leibler (KL) divergence between two LID distributions by Bailey et al. (2022). In Appendix C.5, we have shown that KL divergence is a ‘weak approximation’ of the Fisher-Rao metric on statistical manifolds in the context of LID distributions.
>
> We would like to highlight that Theorems 1 and 3 (by Houle, 2017a) consider the LID for each sample, i.e., each sample has an LID value. Our theory in section 4 makes a novel and important extension to the distance between two LID **distributions** (LIDs of a set of samples). There are fundamental differences between new theoretical results and background definitions of LID.
>
> We would appreciate it if the reviewer could suggest any other related work on LID distributions that we are not aware of. We are happy to compare and cite any relevant works.
>
> ---
>
> **Q2**: Borrowed definition of LID (weakness NO.2 and No.3) and additional contribution (question No.3 by the reviewer).
>
> **A2**: We confirm that Definition 1 and Theorem 1 (in our paper) are defined in section 2.2 of the original paper by Houle, 2017a. Note that Section 3 of our paper  (where Definition 1 and Theorem 1 are located)  is used to provide the necessary background, as a prelude to our new theoretical results in Section 4.
>
> Except for the definitions $LID$ and $IntrDim$ by Houle, 2017a, and the definition of Fisher-Rao (mentioned by the reviewer in question No.3), which are existing theories, all other parts of this paper are novel theoretical contributions. More specifically, Section 3 on pages 3-4 provides background for our new theoretical results in Section 4 from pages 4-6 and corresponding proofs in Appendix C from pages 15-18.
>
> Please kindly find the following summary and clarification of our new theoretical contributions:
> - Lemma 1 and Definition 3 in Sections 4.1 and 4.2 show the Fisher-Rao metric between two LID **distributions** is their absolute log ratio.
> - Definition 4 in Section 4.3, the Fr ́echet variance can be interpreted as the variance of the local intrinsic dimensionalities of the distributions in ${\mathcal F}$, taken in logarithmic scale.
> - Theorem 2 and Corollary 2.1 in Section 4.3 provide three new implications for LID distributions. In short, they show that LID **distributions** should be compared using geometric mean.
> - Theorem 4 in Appendix C.5 shows that the arithmetic means of distributional LIDs are obtained when the asymptotic Fisher-Rao metric is replaced by the asymptotic KL distance in the derivation of the Fr ́echet mean. It also shows that KL distance for two LID distributions is only a ‘weak approximation’ of the Fisher-Rao metric on statistical manifolds.
>
> We confirm that none of the new results above has been shown in Houle, 2017a. That work provided a definition of LID, but did not consider its use for machine learning, nor did it consider how to compare distributions that are characterized using LID.  In contrast, our work shows a principled way to compare distributions, based on their characterization within an asymptotically small neighborhood using LID . In addition to the theory, LDReg regularizes the LIDs, which, to the best of our knowledge, is new from an application perspective. Existing works focused on LID as a descriptive measure, rather than as a component for optimization within a learning process.
>
> We genuinely hope the above explanation can help address your concerns. Please let us know if you’d like further clarification on this aspect.

---

> ### Author Response · Authors · 2023-11-17
> **Response to Reviewer 85fu (2/3)**
>
> **Q3**: Experimental results lack in-depth analysis (weakness No.4), and question No.4 mentioned by the reviewer.
>
> **A3**: For the setting of 100 epoch pre-training with LDReg and transfer learning evaluation, Table 2 indeed shows an underperformance on 2 out of 7 datasets, and Table 3 shows an underperformance using $AP_{75}$ on the object detection task (but not the other 2 metrics). Please allow us to clarify that: 1) This is quite common in SSL evaluation, similar results (underperformance on a few datasets) can also be observed in other popular SSL works (Grill et al., 2020; Zbontar et al., 2021; Bardes et al., 2022). 2) For 1000 epoch pretraining (a more expensive but accurate setting), LDReg demonstrates consistent improvements over the baselines on all datasets and evaluation metrics. It is well known that longer pre-training is beneficial for SSL, and LDReg shows an even larger margin and consistent outperformance on all datasets and metrics under this setting.
>
> We have added new experiments to compare with/without LDReg in 200 epochs pretraining and then combined them with results (100 and 1000 epochs) in the initial submission. It can be observed that the margin of improvement can be increased with longer pretraining and consistent improvements.
>
> | Method | Regularization | Batch Size | Epochs | ImageNet | Food-101 | CIFAR-10 | CIFAR-100 | Birdsnap |   Cars   |    DTD   |
> |:------:|:--------------:|:----------:|:------:|:--------:|:--------:|:--------:|:---------:|:--------:|:--------:|:--------:|
> | SimCLR |        -       |    2048    |   100  |   64.3   |   69.0   |   89.1   |  **71.2** |   32.0   |   36.7   | **67.8** |
> | SimCLR |      LDReg     |    2048    |   100  | **64.8** | **69.1** | **89.2** |    70.6   | **33.4** | **37.3** |   67.7   |
> | SimCLR |        -       |    2048    |   200  |   66.2   |   69.9   |   89.4   |    71.5   | **33.9** |   37.1   |   67.7   |
> | SimCLR |      LDReg     |    2048    |   200  | **66.6** | **70.5** | **90.0** |  **72.2** |   33.4   | **37.9** | **68.9** |
> | SimCLR |        -       |    4096    |  1000  |   69.0   |   71.1   |   90.1   |    71.6   |   37.5   |   35.3   |   70.7   |
> | SimCLR |      LDReg     |    4096    |  1000  | **69.8** | **73.3** | **91.8** |  **75.1** | **38.7** | **41.6** | **70.8** |

---

> ### Author Response · Authors · 2023-11-17
> **Response to Reviewer 85fu (3/3)**
>
> **Q4**: It is unclear how dimensional collapse was observed in the experiment.
>
> **A4**:
> The dimensional collapse means that the representation only intrinsically spans over lower dimensions. There is no fixed threshold in the literature that has been used to determine the existence of "collapse". Existing works (Jing et al., 2022; Garrido et al., 2023a) suggest SSL should avoid dimension collapse. Hence, a higher intrinsic dimension is better. Effective rank has been used as a global measure for evaluating the intrinsic dimensions. In Table 1, it can be observed that LDReg can increase the effective rank of the representation, thus improving the representation qualities, which is evaluated using linear probing in Table 1.
>
> The existing work (Jing et al., 2022) also shows that the extreme case of dimension collapse is the complete (mode) collapse, where the output is a constant vector. What degree (or threshold) corresponds to the notion of dimension collapse is yet to be defined in the literature.
>
> The theory we developed for LID distributions can also influence the global intrinsic dimension to become lower (the opposite of LDReg). Instead of maximizing the LID to become distributionally higher, we can also minimize the LID to become distributionally lower, denoted as **Min LID**. Technical details are in Appendix G.2, and we copy-pasted the result here. All results are based on 100 epoch pretraining with SimCLR and ResNet-50.
>
> | Method | Regularization | $\beta$ |     Linear Acc    | Effective Rank | Geometric mean of LID |
> |:------:|:--------------:|:-------:|:-----------------:|:--------------:|:---------------------:|
> | SimCLR |      LDReg     |   0.01  |       **64.8**       |      529.6     |         20.04         |
> |        |        -       |    -    |        64.3       |      470.2     |         18.79         |
> |        |     Min LID    |   0.01  |        64.2       |      150.7     |         16.01         |
> |        |     Min LID    |   0.1   |        63.1       |      15.0      |          3.81         |
> |        |     Min LID    |   1.0   |        46.4       |       1.0      |          1.64         |
> |        |     Min LID    |   10.0  | Complete collapse |        -       |           -           |
>
> Note that MinLID denotes deliberately making LID become distributionally lower, and LDReg denotes making LID become distributionally higher.
>
> It can be observed that as effective rank decreases, it triggers a complete (mode) collapse at the extreme level. It can also be observed that as effective rank decreases, the performance also decreases. Similar results can be observed with LID. Preventing the dimension collapse problem could be seen as avoiding span over lower dimensions.
>
> ---
>
> **Q5:** Typos in definition 1 and page 19.
>
> **A5:** We have fixed the typos in the revision.
>
> ---
>
> **Q6**: What ‘Pr’ denotes.
>
> **A6**: $Pr$ denotes probability. $F(r)$ denotes the cumulative distribution function (CDF) of the local distance distribution, which is the probability of the sample distance lying within a threshold $r$ — that is, $F(r) \triangleq Pr[d(\mathbf{x},\mathbf{s})\leq r]$.
>
> ---
> **Q7**: Surface of the sphere ball
>
> **A7**:
> The regularization objective with respect to each query sample q in a batch is to
> Minimize $-log(ID)$
> $=-log(\mu_k/(w_k-\mu_k) )$
> $=-log(1/(w_k/\mu_k  - 1))$
> $=log(w_k/\mu_k - 1)$
>
> Here, we have a ball centered at a query sample q with radius of the ball equal to $w_k$ and $\mu_k$ is the average distance to q’s nearest neighbor within the ball.
>
> The objective encourages the term $w_k/\mu_k$ to be low, or equivalently $\mu_k/w_k$ to be high.   Since it is guaranteed that $\mu_k<=w_k$, this regularization encourages the “normalized mean” $\mu_k/w_k$ to be high.   I.e. In the local context of query q, the nearest neighbors of q will be pushed towards the edge (surface) of the ball, leaving more “empty space” inside the ball.  Such an empty space phenomenon is a well-known phenomenon associated with the presence of higher dimensionality.  So roughly speaking, the regularization encourages the local ball around each sample to have more empty space (higher local dimensionality).
>
> ---
>
> We do hope this addresses the reviewer’s question.  Please let us know if further clarification is needed.

---

> ### Author Response · Authors · 2023-11-21
> **Thanks for your feedback**
>
> Dear reviewer 85fu,
>
>
> Thank you very much for your initial comments. Hope our clarifications can help address some of your concerns. Please kindly let us know if you have any additional questions. We will try our best to answer them before the rebuttal ends.

---

> ### Comment · Reviewer_85fu · 2023-11-22
>
> Thank you very much for the thoughtful response. These replies have addressed my concerns and further demonstrated the excellent performance of this work.

---

> > ### Author Response · Authors · 2023-11-23
> > **Thank you**
> >
> > Dear reviewer 85fu,
> >
> > Thank you very much for your encouraging feedback.

---

### Official Review · Reviewer_VyQK · 2023-10-31

**Soundness:** 3 good
**Presentation:** 3 good
**Contribution:** 3 good
**Rating:** 6
**Confidence:** 4

**Summary:**

This paper presents a novel variant of the Fisher-Rao metric, and then proposes a local dimensionality regularization (LDReg) to alleviate the dimension collapse in self-supervised learning (SSL). Moreover, it is verified that geometric mean is suitable to manipulate the intrinsic dimension, and therein LDReg maximizes the logarithm of the geometric mean of the sample-wise LID to have nonuniform local nearest neighbor distance distributions. Empirical evaluations are provided to demonstrate the effectiveness of the proposed approach in some degree.

**Strengths:**

+ It is interesting to take the perspective of the local intrinsic dimension to remedy the dimensionality collapse.
+ Applying the geometric mean of the local intrinsic dimensionality (LID) as a regularizer for SSL is novel.

**Weaknesses:**

- The empirical evaluations look insignificant. In Table 1 and 3, the performance improvements are relatively weak. The reviewer is curious that these improvements are significant or not?

- The reviewer is curious about the sensitivity to the locality parameter. What about the performance of the LID based regularizer with respect to the locality parameter $k$? Is it sensitive to the parameter $k$? There is another parameter $N$. Is it the number of samples of the dataset, or is it the batch size? If the later case, what about the sensitivity of the performance with respect to the batch size (or the density of the samples)?


- $F$ is assumed to be differentialble at $r$. However, if $F(r)$ is defined as the prob of the sample distance lying within a threshold $r$, how can we define the differential at $r$ for $F(r)$? Moreover, $LID_F*$ defined as the limit of $LID_F(r)$ when $r\rightarrow 0$, is refered as LID. How about the approximation quality of such a local intrisic dimension estimator? In another words, is it a good estimator of the local intrisinc dimension? Is any theoretical or empirical evidence to show the quality of the so-defined LID to estimate the local intrinsic dimension of the data?

**Questions:**

- Since that the results in Table 1 and 3, the improvements are relatively weak. Are these improvements significant? Or it is from other minor modification or some random fluctuation?

- What about the performance of the LID based regularizer with respect to the parameter $k$? Is the parameter $k$ affected by the batch size (or the density of the samples)?

- Since that $LID_F*$ defined as the limit of $LID_F(r)$ when $r\rightarrow 0$. How about the approximation quality of such a local intrisic dimension estimator? In another words, is it a good estimator of the local intrisinc dimension? Is any theoretical or empirical evidence to show the quality of the so-defined LID to estimate the local intrinsic dimension of the data?

---

> ### Author Response · Authors · 2023-11-17
> **Response to Reviewer VyQK (1/2)**
>
> Thank you very much for reviewing our paper and the valuable comments. We have prepared a clarification for each of your questions, please kindly let us know if anything is still unclear.
>
> ---
>
> **Q1**: Empirical improvements (weaknesses No.1 and question No.1).
>
> **A1**:
> We believe the improvements are *substantial* viewed in the context of SSL research. The improvements brought by the current state of the art methods are mostly within 0.5-2% [1, 2]. We have run a new set of experiments with one more SOTA SSL method, MoCo [3], in a 100-epoch pretraining setting with ResNet. The results in the table below highlights a consistent ~1% improvement.
>
> | Method | Regularization | Locality parameter (k)  | Linear Evaluations |
> |:------:|:--------------:|:-----------------------:|:------------------:|
> |  MoCo  |        -       |            -            |        68.7        |
> |  MoCo  |      LDReg     |            32           |      69.5      |
> |  MoCo  |      LDReg     |            64           |      69.5      |
> |  MoCo  |      LDReg     |           128          |      **69.6**      |
>
> Due to the high computational cost, we did not extensively search the locality hyperparameters except for SimCLR, nor the hyperparameters of each model in the downstream tasks (Tables 2 and 3). For the baseline methods, we adopted the hyperparameters suggested by a popular SSL library (see Appendix F for details) and these hyperparameters were not tuned for our LDReg. It is possible that with a fine-grained hyperparameter search,  LDReg could achieve further improved performances.
>
> We don't believe the improvements are due to random fluctuations. All our experiments were run with the same random seed with/without LDReg. We also have experiments running with different $k$ and $\beta$ hyperparameters (Figure 5 in Appendix G.3), which all use the same random seed. The consistent improvement of our LDReg different SSL methods is clearly shown in Tables 1 - 3 and Figure 5 (a-b).
>
> We have ensured that, for each pair of comparisons and each SSL method, the only difference is with/without LDReg. All other experimental settings are the same, such as random seed, augmentations, architectures, implementation of the main loss (SSL objective) function, software version, and GPU model/architectures. It is very unlikely that this consistent improvement is due to random fluctuation.
>
> We performed a McNemar's test on SimCLR v.s. SimCLR+LDReg. The null hypothesis is that the linear evaluation performance of the two models is equal. Based on the contingency table obtained by running predictions on the ImageNet validation set. The McNemar Test produces a p-value of 0.0041. Considering a significance level of 0.05, we can reject the null hypothesis.
>
> |                  | SimCLR+LDReg Correct  | SimCLR+LDReg Incorrect |
> |:----------------:|:---------------------:|:----------------------:|
> |  SimCLR Correct  |         28668         |          3467          |
> | SimCLR Incorrect |          3711         |          14154         |
>
> ---
> [1] Bardes, Adrien, Jean Ponce, and Yann LeCun. "VICReg: Variance-Invariance-Covariance Regularization for Self-Supervised Learning." ICLR. 2022.
>
> [2] Zbontar, Jure, et al. "Barlow twins: Self-supervised learning via redundancy reduction." ICML. 2021.
>
> [3] He, Kaiming, et al. "Momentum contrast for unsupervised visual representation learning." CVPR. 2020.

---

> ### Author Response · Authors · 2023-11-17
> **Response to Reviewer VyQK (2/2)**
>
> **Q2**: Sensitivity to locality parameters  (weaknesses No.2 and question No.2).
>
> **A2**:
> In Appendix G.3, we have evaluated the sensitivity to the $k$ and $\beta$. It can be observed that LDReg can consistently improve the SimCLR for $k$ in the range of 32 to 256. LDReg is relatively stable for $\beta$ in the range of 0.1 to 0.001. Also, in our newly added experiments in A1, the performance is consistently improved for $k$ in the range of 32 to 128 for MoCo. For convenience of discussion, we have copy-pasted the results to the table below, where "-" denotes the baseline without using LDReg.
>
> For sensitivity to $k$ with $\beta=0.01$:
> | Regularization |  k  | Linear Acc |
> |:--------------:|:---:|:----------:|
> |        -       |  -  |    64.3    |
> |      LDReg     |  32 |    64.3    |
> |      LDReg     |  64 |    **64.8**    |
> |      LDReg     | 128 |    64.6    |
> |      LDReg     | 256 |    **64.8**    |
>
> For sensitivity to $\beta$ with $k=128$:
> | Regularization | $\beta$ | Linear Acc |
> |:--------------:|:-------:|:----------:|
> |        -       |    -    |    64.3    |
> |      LDReg     |   0.1   |    64.3    |
> |      LDReg     |   0.01  |    **64.6**    |
> |      LDReg     |  0.001  |    **64.6**    |
>
> The $N$ refers to the effective batch size. We have added a new experiment to test the sensitivity to different batch sizes with SimCLR. We reduce the $k$ at the same rate as $N$ is reduced, e.g. 16 and 32 for batch sizes 512 and 1024, respectively. The results are in the table below. It can be observed that LDReg can consistently improve the baseline in different batch sizes.
>
> | Method | Regularization |   N=512  |  N=1024  |  N=2048  |
> |:------:|:--------------:|:--------:|:--------:|:--------:|
> | SimCLR |        -       |   63.6   |   64.2   |   64.3   |
> | SimCLR |      LDReg     | **64.1** | **64.7** | **64.8** |
>
>
>
> ---
> **Q3**: LID estimation quality (weaknesses No.3 and question No.3) and the so-defined LID.
>
> **A3**:
> $ LID_{F}^{*} $ is theoretically defined as the limit of $ LID_{F}(r) $ as the distance $r$ tends to $0$.
>
> In practice, $LID_{F}(r)$ is unavailable, and it is $ LID_{F}^{*} $ that we are interested in as the “LID”. LID needs to be estimated using a sample and there exist different possible estimators for this task.   The method of moments estimator, which we use, has previously been studied in [4, 5] and has found to be quite effective in terms of mean squared error and is close in performance to the performance maximum likelihood estimator.  For our purposes, method of moments estimator is convenient to use, due to its computational simplicity (being somewhat simpler than the maximum likelihood estimator) and for this reason we adopt it.    Other estimators might alternatively be used.   Note that our work in this paper is not about developing a new LID estimation method.  Rather, we leverage an existing estimation method for developing a novel regularization.
>
> Below, we compare MoM and maximum likelihood estimation (MLE) in the context of SSL. The following result shows that MoM and MLE produce similar LID estimations.
>
> | Estimator | Geometric mean of LID |
> |:---------:|:---------------------:|
> |    MoM    |         18.08         |
> |    MLE    |         18.44         |
>
> ---
>
> [4] Amsaleg, Laurent, et al. "Extreme-value-theoretic estimation of local intrinsic dimensionality." Data Mining and Knowledge Discovery 32.6 (2018): 1768-1805.
>
> [5] Amsaleg, Laurent, et al. "Estimating local intrinsic dimensionality." Proceedings of the 21th ACM SIGKDD International Conference on Knowledge Discovery and Data Mining. 2015.

---

> ### Author Response · Authors · 2023-11-21
> **A follow up message**
>
> Dear reviewer VyQK,
>
> Thanks again for reviewing our paper. Please let us know if you have any additional questions or require further clarification. We are happy to address them before the rebuttal ends.

---

> > ### Comment · Reviewer_VyQK · 2023-11-22
> >
> > The reviewer appreciates the careful clarification in the rebuttal and great effort in providing more experimental supports. These responses have by and large resolved the reviewers concerns. Thus, the reviewer would like to increase the rating slightly.

---

> > > ### Author Response · Authors · 2023-11-23
> > > **Thank you**
> > >
> > > Dear reviewer VyQK,
> > >
> > > Thanks very much for your recognition of our work and the encouraging comments. They have greatly improved our work.

---

### Official Review · Reviewer_LKEn · 2023-11-01

**Soundness:** 3 good
**Presentation:** 3 good
**Contribution:** 2 fair
**Rating:** 5
**Confidence:** 3

**Summary:**

[I made a mistake in the form.] I found out that I have accidentally checked the "First Time Reviewer" question, but in fact, I'm not. It seems that I cannot undo it now, so I'm instead writing it here.

This paper proposes a new regularization technique for self-supervised learning. There are many recent works on preserving the internal diversity of self-supervised representation, and one approach is to preserve the effective dimensionality of the representation. Unlike existing work that tries to preserve global dimensionality, this paper argues that local dimensionality might still collapse. To resolve this issue, the paper proposes the local intrinsic dimensionality (LID). Using the Fisher-Rao metric on LID, the proposed method adds a regularization term that makes the distribution of the representation far from the most simplistic distribution (with dimensionality one). Experiments show that the proposed method indeed improves the performance.

**Strengths:**

- The motivation behind the proposed method (local dimensionality collapse) makes sense, and it is demonstrated empirically.

- The proposed regularization is elaborately designed based on well-founded theories (LID representation, Fisher-Rao metric).

- Experiments show that the local dimensionality indeed improves. The performance improvement itself is somewhat incremental except for a few cases. However, considering the recent self-supervised learning works, this is understandable.

**Weaknesses:**

- There is no comparison to other similar methods for improving self-supervised learning. In particular, there is no comparison to ones with global dimensionality regularization (which has close relationships with the proposed method), even though they were mentioned in the paper.

- The proposed method requires the calculation of distance distributions during training. This can be somewhat heavy, depending on the actual settings. Ideally, self-supervised learning is meant to be performed on large-scale data, so this point can be even more burdensome. How big is the actual computational burden? I said that the incremental performance improvement is understandable, but it might not be really beneficial if the computational burden is quite high, considering there are also other recent alternatives.

**Questions:**

Please see the above weaknesses.

---

> ### Author Response · Authors · 2023-11-17
> **Response to Reviewer LKEn**
>
> Thank you very much for reviewing our paper and the valuable comments. Please find our response to your questions below:
>
> ---
>
> **Q1**: Comparing to decorrelating features
>
> **A1**:
> Thanks for your suggestions. Please find the results for applying LDReg on these methods below.
>
> |    Method    | Regularizer | Linear Evaluation Accuracy | Effective Rank | Geometric mean of LIDs |
> |:------------:|:-----------:|:--------------------------:|:--------------:|:----------------------:|
> |     BYOL     |      -      |            67.6            |      583.8     |          15.9          |
> |     BYOL     |    LDReg    |          **68.5**          |      594.0     |          22.3          |
> |    VICReg    |      -      |            66.7            |      546.7     |          21.5          |
> |    VICReg    |    LDReg    |          **66.9**          |      602.4     |          22.5          |
> | Barlow Twins |      -      |            65.5            |      602.1     |          20.8          |
> | Barlow Twins |    LDReg    |          **65.6**          |      754.0     |          24.1          |
>
> In our initial submission, we focused on addressing the dimensional collapse issue of methods that are known for being susceptible to such issues. VICReg and Barlow Twins are SSL methods rather than regularizers. Compared to BYOL, VICReg and Barlow Twins use different projector architectures and loss functions. It’s not fair to compare LDReg across different types of SSL methods. For example, BYOL with LDReg has a linear evaluation of 68.5, yet BYOL alone can achieve 67.6, while VICReg under the same setting (100 epochs of pretraining) is only 66.7. However, if we apply our regularizer to these methods, their performance can all be improved, as shown in the above table.
>
> To fairly compare the regularizers, we use the covariance (denoted as Cov) and variance (denoted as Var) as alternative regularizers to replace LDReg and apply to BYOL. Note that they are pseudo-global dimension regularizers, as we cannot use the entire training set to calculate the covariance, it is calculated on a mini-batch. We also performed a hyperparameter search for Cov and Var ($\beta$ for the strength of the regularization). We used the same regularization formula as VICReg for Cov and Var.
>
> $C(Z) = \frac{1}{n - 1} \sum_{i=1}^{n} (z_{i} - \bar{z})(z_{i} - \bar{z})^{T}$, $\bar{z} = \frac{1}{n} \sum_{i=1}^{n} z_{i}$,
> $Cov(Z) = c(Z) = \frac{1}{d} \sum_{i \ne j} [C(Z)]_{i,j}^2,  $ where $d$ is the representation dimension.
>
> $Var(Z) = \frac{1}{d} \sum_{j=1}^{d} \max(0, \gamma - S(z^{j}, \epsilon))$ where $S(\cdot)$ is the standard deviation.
>
> For Cov, we regularize its off-diagonal coefficients in the same manner as was done for VICReg. We apply the regularization on representations learned by the encoder, the same as in LDReg. The results can be found in the table below. All results are based on 100 epoch pretraining with BYOL and ResNet-50. All experimental settings are exactly the same as LDReg except for the regularization term.
>
> | Method | Regularizer | $\beta$ | Linear Evaluation Accuracy | Effective Rank | Geometric mean of LID |
> |:------:|:-----------:|:-------:|:--------------------------:|:--------------:|:---------------------:|
> |  BYOL  |     None    |    -    |            67.6            |      583.8     |          15.9         |
> |  BYOL  |     Cov     |   0.01  |            67.6            |      583.5     |          15.9         |
> |  BYOL  |     Cov     |   0.1   |            67.5            |      593.5     |          15.8         |
> |  BYOL  |   Cov+Var   |   0.01  |            67.8            |      539.2     |          15.5         |
> |  BYOL  |   Cov+Var   |   0.1   |            67.7            |      798.4     |          16.8         |
> |  BYOL  |    LDReg    |  0.005  |          **68.5**          |      594.0     |          22.3         |
>
> Although the covariance and variance regularizer (Cov+Var) can increase the global dimension, it does not improve the local dimension. It also has a rather minor effect (<=0.2%) on the linear evaluation accuracy, whereas LDReg improves by almost 1%. This further confirms the effectiveness of LDReg.
>
> ---
> **Q2**: Computational burden
>
> **A2**: In Table 4 (Appendix F), we provided a wall-clock comparison, which shows that in the 100 epoch pretraining setting, the training time is similar with/without LDReg, i.e., ~27 hours for pretraining with 4 GPUs in a distributed setting. This is because the pairwise distance calculation in LDReg is performed within a minibatch via matrix multiplication, which can be efficiently done without incurring a heavy computation burden.

---

> ### Author Response · Authors · 2023-11-21
> **Any additional questions?**
>
> Dear reviewer LKEn,
>
> Please let us know if you have any additional questions or require further clarification. We are happy to address them before the rebuttal ends.

---

> ### Comment · Reviewer_LKEn · 2023-11-22
>
> Thank you for the detailed answers. I'm also convinced by the new experiments, showing that LDReg shows superb performance.

---

> ### Author Response · Authors · 2023-11-23
> **Thank you!**
>
> Dear reviewer LKEn,
>
> Thank you very much for your encouraging feedback. We would really appreciate it if you could also update the rating. This would mean a lot to us. Thank you very much!

---

### Official Review · Reviewer_oLx3 · 2023-11-01

**Soundness:** 4 excellent
**Presentation:** 4 excellent
**Contribution:** 2 fair
**Rating:** 8
**Confidence:** 4

**Summary:**

The authors propose to increase the accuracy of deep models by regularizing the local intrinsic dimensionality (LID) of features. They observe that without the proposed regularization, the dimensionality collapses locally, even if the global dimension remains constant.

**Strengths:**

(1) The paper is well presented and well supported by theory.
(2) The method seems intuitive.

**Weaknesses:**

(1) It's hard to reference an equation without equation numbers. In your overall optimization objective, it is unclear how $LID^*_F$ is calculated. I see in section 3 that this quantity is the result of a limit of some other quantity $LID_F$, which then depends on differentiating a function. It would be hard for me to implement this loss function just from reading this paper. How are all of these values calculated?

(2) Follow up from (1): It is unclear how you estimate a "local" dimensionality from a mini-batch of samples. The mini-batch is sampled over the entire dataset, so none of them lie in the local neighborhood of other samples within the batch. I don't think this is addressed.

(3) How does the proposed LIDs regularizer compare to regularizing global dimensionality by decorrelating features. The authors site a few works at the end of the first paragraph of the intro, but do not compare against them. For instance [Barlow twins] and [VICE-Reg] are popular ways of doing this.

(4) Follow up from (3): Regularizing global dimensionality makes sense to me, but regularizing local dimensionality does not. e.g. looking at Figure 1(c), the authors show a few examples of local dimensionality collapse compared with one example of high local dimensionality. To me, it looks like the examples with low local dimensionality (i.e. low LID, but constant GID) exhibit more structure and therefore could be better features. How do you expect to learn good features when you regularize the distribution to be a random gaussian both globally and locally?

(5) I don't see any results showing a correlation between test accuracy and LID. I would expect to see a plot showing that when LID increases, so does accuracy. Perhaps I missed it, if so please point me there. Furthermore, I would expect GID and LID to be correlated; so I would expect some result that shows improving LID with constant GID improves accuracy. Perhaps the LID and GID scores could be added to Table 1?

**Questions:**

See above.

---

> ### Author Response · Authors · 2023-11-17
> **Response to Reviewer oLx3 (1/3)**
>
> Thanks for your insightful reviews. Please find our response to your questions below:
>
> ---
> **Q1:** Optimization objective calculation and implementation
>
> **A1:** We estimate the LID for each sample using the method of moments (MoM) estimator [1] due to its simplicity. Other estimators, such as maximum likelihood estimation (MLE), would also be possible.
>
> The estimation requires a batch of reference samples; for simplicity, we use the online mini-batch for this purpose. For each sample, MoM calculates the pairwise distances between all reference samples. The LID of a particular sample can then be estimated using MoM as $ID^{*}= - \frac{\mu_k}{\mu_k - w^{k}}$, where $k$ is a hyperparameter denoting the number of nearest neighbors, $w^{k}$ is the distance from the sample to its k-th nearest neighbor and $\mu_k$ is the average distance from the sample to its k nearest neighbors. We estimate the LID for all samples in the batch and regularize it on the representations learned by the encoder.
>
> The Pseudocode of the implementation is available in Appendix J. The Code is available in the anonymous link of the Reproducibility Statement and will be open-source after review.
>
> Thanks for your suggestion, we have added equation numbers in the updated draft.
>
> ---
> **Q2:** Mini-batch and local neighborhoods.
>
> **A2:** As answered in A1, we calculate the distance from a sample to all other samples within the mini-batch to find the nearest neighbors. These nearest neighbors in the mini-batch might not be the "closest" if we consider the whole dataset as reference points, but we find that using a mini-batch for estimation is reasonably effective in the deep representation space.
>
> We have conducted a new estimation experiment using the ImageNet test set and representation learned by SimCLR. We randomly sample a batch of samples and then compare the LID values estimated on the entire dataset (50,000 samples) vs. within the batch (2048 samples). The experiment was repeated 5 times with different random batches, and the results are shown below.
>
> SimCLR:
> | Reference Points | LID geometric mean |
> |:----------------:|:------------------:|
> |  Entire Dataset  |    18.18 ± 0.31    |
> |    Mini-batch    |    21.19 ± 0.24    |
>
>
> SimCLR with LDReg:
> | Reference Points | LID geometric mean |
> |:----------------:|:------------------:|
> |  Entire Dataset  |    19.89  ± 0.27    |
> |    Mini-batch    |    23.94 ± 0.25    |
>
> The Mini-batch estimation is slightly higher than using the entire dataset as reference points. However, the difference is relatively small, considering the scale of the entire training set. Please note that it is infeasible to use the entire dataset to estimate LID during training due to efficiency and memory reasons. One might also use a moving queue like MoCo [2] to expand the number of reference points.
>
> Using a mini-batch to estimate LID has been used in existing machine-learning applications, and it achieved solid results in adversarial attack detections [3], understanding deep learning [4, 5], and robust learning with noisy labels [6].
>
> ---
>
> [1] Amsaleg, Laurent, et al. "Extreme-value-theoretic estimation of local intrinsic dimensionality." Data Mining and Knowledge Discovery 32.6 (2018): 1768-1805.\
> [2] He, Kaiming, et al. "Momentum contrast for unsupervised visual representation learning." CVPR, 2020.\
> [3] Ma, Xingjun, et al. "Characterizing adversarial subspaces using local intrinsic dimensionality." ICLR 2018.\
> [4] Ansuini, Alessio, et al. "Intrinsic dimension of data representations in deep neural networks." NeurIPS 2019.\
> [5] Pope, Phil, et al. "The Intrinsic Dimension of Images and Its Impact on Learning." ICLR. 2020.\
> [6] Ma, Xingjun, et al. "Dimensionality-driven learning with noisy labels." ICML, 2018.

---

> ### Author Response · Authors · 2023-11-17
> **Response to Reviewer oLx3 (2/3)**
>
> **Q3**: Comparing to decorrelating features
>
> **A3**:
> Thanks for your suggestions. Please find the results for applying LDReg on these methods below.
>
> |    Method    | Regularizer | Linear Evaluation Accuracy | Effective Rank | Geometric mean of LIDs |
> |:------------:|:-----------:|:--------------------------:|:--------------:|:----------------------:|
> |     BYOL     |      -      |            67.6            |      583.8     |          15.9          |
> |     BYOL     |    LDReg    |          **68.5**          |      594.0     |          22.3          |
> |    VICReg    |      -      |            66.7            |      546.7     |          21.5          |
> |    VICReg    |    LDReg    |          **66.9**          |      602.4     |          22.5          |
> | Barlow Twins |      -      |            65.5            |      602.1     |          20.8          |
> | Barlow Twins |    LDReg    |          **65.6**          |      754.0     |          24.1          |
>
> In our initial submission, we focused on addressing the dimensional collapse issue of methods that are known for being susceptible to such issues. VICReg and Barlow Twins are SSL methods rather than regularizers. Compared to BYOL, VICReg and Barlow Twins use different projector architectures and loss functions. It’s not fair to compare LDReg across different types of SSL methods. For example, BYOL with LDReg has a linear evaluation of 68.5, yet BYOL alone can achieve 67.6, while VICReg under the same setting (100 epochs of pretraining) is only 66.7. However, if we apply our regularizer to these methods, their performance can all be improved, as shown in the above table.
>
> To fairly compare the regularizers, we use the covariance (denoted as Cov) and variance (denoted as Var) as alternative regularizers to replace LDReg and apply to BYOL. Note that they are pseudo-global dimension regularizers, as we cannot use the entire training set to calculate the covariance, it is calculated on a mini-batch. We also performed a hyperparameter search for Cov and Var ($\beta$ for the strength of the regularization). We used the same regularization formula as VICReg for Cov and Var.
>
> $C(Z) = \frac{1}{n - 1} \sum_{i=1}^{n} (z_{i} - \bar{z})(z_{i} - \bar{z})^{T}$, $\bar{z} = \frac{1}{n} \sum_{i=1}^{n} z_{i}$,
> $Cov(Z) = c(Z) = \frac{1}{d} \sum_{i \ne j} [C(Z)]_{i,j}^2,  $ where $d$ is the representation dimension.
>
> $Var(Z) = \frac{1}{d} \sum_{j=1}^{d} \max(0, \gamma - S(z^{j}, \epsilon))$ where $S(\cdot)$ is the standard deviation.
>
> For Cov, we regularize its off-diagonal coefficients in the same manner as was done for VICReg. We apply the regularization on representations learned by the encoder, the same as in LDReg. The results can be found in the table below. All results are based on 100 epoch pretraining with BYOL and ResNet-50. All experimental settings are exactly the same as LDReg except for the regularization term.
>
> | Method | Regularizer | $\beta$ | Linear Evaluation Accuracy | Effective Rank | Geometric mean of LID |
> |:------:|:-----------:|:-------:|:--------------------------:|:--------------:|:---------------------:|
> |  BYOL  |     None    |    -    |            67.6            |      583.8     |          15.9         |
> |  BYOL  |     Cov     |   0.01  |            67.6            |      583.5     |          15.9         |
> |  BYOL  |     Cov     |   0.1   |            67.5            |      593.5     |          15.8         |
> |  BYOL  |   Cov+Var   |   0.01  |            67.8            |      539.2     |          15.5         |
> |  BYOL  |   Cov+Var   |   0.1   |            67.7            |      798.4     |          16.8         |
> |  BYOL  |    LDReg    |  0.005  |          **68.5**          |      594.0     |          22.3         |
>
> Although the covariance and variance regularizer (Cov+Var) can increase the global dimension, it does not improve the local dimension. It also has a rather minor effect (<=0.2%) on the linear evaluation accuracy, whereas LDReg improves by almost 1%. This further confirms the effectiveness of LDReg.

---

> ### Author Response · Authors · 2023-11-17
> **Response to Reviewer oLx3 (3/3)**
>
> **Q4**: Figure 1(c) and low LID
>
> **A4**:
> We would like to clarify that the purpose of the example is to illustrate that it is possible to have datasets that have almost the same global dimension, but rather different local dimension characteristics. We generated this dataset synthetically without using any SSL objective. We agree with the reviewer that it is perhaps surprising that the local dimensionality needs to be sufficiently high for effective learning, but indeed this was what we have discovered empirically and it serves as the motivation for the local regularization that we formulate.
>
> We have revised the paper and added figures (Figure 4 in Appendix G.2, page 23) with t-SNE plots for SSL-learned representations. It shows that when the LIDs are low, the "local structure" is clearly not helpful for the representation.
>
> Comparing Figure 4(b) without any regularizations, Figure 4(d) shows the minimized LID distributions, and the representation has an effective rank of 1. This significantly reduced the accuracy to 46.4, as shown in Table 13. From Figure 4(d), it can be observed that representations only span over a line, making data points hard to separate from others. Similar results can be found in Figure 4(c), which have an effective rank equal to 15. Compared with LDReg shown in Figure 4(a), it shows a better separability for classes labelled as blue.
>
> ---
>
> **Q5**: Correlation between test accuracy and LID
>
> **A5**: Thanks for your suggestion. We have now reported the geometric mean of LIDs in Table 1 and Table 13 in Appendix G.2 (and the table below). It shows that, for the same SSL method and model, LID is well correlated with global intrinsic dimensions (GID) and the linear evaluation performance, i.e., higher LID distribution results in higher GID. However, we find it hard to produce the scenario you mentioned to increase LID while keeping GID constant.
> The ‘Min LID’ in the table below means deliberately making LID lower following our theoretical results.
>
> | Method | Regularization | $\beta$ |     Linear Acc    | Effective Rank | Geometric mean of LID |
> |:------:|:--------------:|:-------:|:-----------------:|:--------------:|:---------------------:|
> | SimCLR |      LDReg     |   0.01  |      **64.8**     |      529.6     |         20.04         |
> | SimCLR |        -       |    -    |        64.3       |      470.2     |         18.79         |
> | SimCLR |     Min LID    |   0.01  |        64.2       |      150.7     |         16.01         |
> | SimCLR |     Min LID    |   0.1   |        63.1       |      15.0      |          3.81         |
> | SimCLR |     Min LID    |   1.0   |        46.4       |       1.0      |          1.64         |
> | SimCLR |     Min LID    |   10.0  | Complete collapse |        -       |           -           |
>
> A similar pattern between LID, effective rank and linear probing accuracy can also be observed in Figure 2(d) and Table 1.

---

> ### Comment · Reviewer_oLx3 · 2023-11-20
> **Response to response**
>
> I thank the authors for their rebuttal responses. I find the responses to be convincing and detailed.
>
> (1) The method is reproducible. Even if it's not entirely clear from main paper how to implement, practitioners can look at the Appendix and code.
>
> (2) Authors show that LID geometric mean on mini-batch and entire dataset is correlated.
>
> (3) The new comparisons to current SoTA regularizers for self-sup. learning are sufficient.
>
> (4) I think the authors do as good a job as possible addressing the intuitive aspect of their work. However, to be completely frank, I suspect many readers from the self-supervised literature will find this work to be counter-intuitive and too "mathy"; this may limit the impact of this work.
>
> (5) The new results showing correlation between LID and accuracy are convincing.
>
> Overall a solid paper.

---

> > ### Author Response · Authors · 2023-11-21
> >
> > Thank you very much for the positive feedback and suggestions. Your quality review is much appreciated.

---

### Author Response · Authors · 2023-11-17
**Summary of changes in the revision**

We made the following changes to the draft based on reviewers' suggestions.

- We added the geometric mean of LID to Table 1. These results were shown in Figure 2(d) in the initial submission. We also added the geometric mean of LID to Table 13 in Appendix G.2. It can be observed that for the same method and model architectures, the geometric mean of LID is correlated with effective rank and linear probing accuracy. This change is to address the concerns of reviewers oLx3 and 85fu.
- In Appendix G.2., Figure 4, we added t-SNE visualizations for learned representations for high and low LID distributions. It can be observed that low LID distributions degrade representation quality, which can be confirmed by Table 13. This change is to address the concern of reviewer oLx3.
- In Appendix G.4, Table 15, we added new experiment results on applying LDReg with global dimension methods (VICReg and Barlow Twins) and MoCo, where consistent improvement can be observed. Technical details can be found in the updated draft (Appendix G.4), and results are copied and pasted here for discussion. This change is to address the concerns of reviewers oLx3, LKEn and VyQK.
- In Table 16, Appendix G.4, we added new experiment results for comparison with decorrelating features as the regularization terms. This change is to address the concerns of reviewers oLx3 and LKEn.
- We have added equation numbers mentioned by reviewer oLx3. We fixed a few typos mentioned by reviewer 85fu.

---

Reviewers oLx3 and LKEn have raised a common question regarding comparing with global dimension regularization (decorrelating features) methods. To address the concerns, we have added new experiments to compare with these methods. We consider two situations. 1) Apply LDReg to these methods, 2) Replace LDReg with decorrelating features regularizers on BYOL.  All experimental settings are the same as in the initial submission. We also report the results of BYOL from the initial submission here for comparison. Results applying LDReg on VICReg and Barlow Twins with ResNet-50 for 100 epochs pretraining can be found below.

|    Method    | Regularizer | Linear Evaluation Accuracy | Effective Rank | Geometric mean of LIDs |
|:------------:|:-----------:|:--------------------------:|:--------------:|:----------------------:|
|     BYOL     |      -      |            67.6            |      583.8     |          15.9          |
|     BYOL     |    LDReg    |          **68.5**          |      594.0     |          22.3          |
|    VICReg    |      -      |            66.7            |      546.7     |          21.5          |
|    VICReg    |    LDReg    |          **66.9**          |      602.4     |          22.5          |
| Barlow Twins |      -      |            65.5            |      602.1     |          20.8          |
| Barlow Twins |    LDReg    |          **65.6**          |      754.0     |          24.1          |

VICReg and Barlow Twins are SSL methods rather than regularizers. Compared to BYOL, VICReg and Barlow Twins use different projector architectures and loss functions. It’s not meaningful to compare LDReg across different types of SSL methods. For example, BYOL with LDReg has a linear evaluation of 68.5, yet BYOL alone can achieve 67.6, while VICReg under the same setting (100 epochs of pretraining) is only 66.7. However, if we apply our regularizer to these methods, their performance can all be improved, as shown in the above table.

---

> ### Author Response · Authors · 2023-11-17
>
> To fairly compare the regularizers, we also added new experiments that replace the LDReg term with global covariance (denoted as Cov) and variance (denoted as Var) as alternative regularizers to replace LDReg. We used the same regularization formula as VICReg for Cov and Var.
>
> $C(Z) = \frac{1}{n - 1} \sum_{i=1}^{n} (z_{i} - \bar{z})(z_{i} - \bar{z})^{T}$, $\bar{z} = \frac{1}{n} \sum_{i=1}^{n} z_{i}$,
> $Cov(Z) = c(Z) = \frac{1}{d} \sum_{i \ne j} [C(Z)]_{i,j}^2,  $ where $d$ is the representation dimension.
>
> $Var(Z) = \frac{1}{d} \sum_{j=1}^{d} \max(0, \gamma - S(z^{j}, \epsilon))$ where $S(\cdot)$ is the standard deviation.
>
> For Cov, we regularize its off-diagonal coefficients in the same manner as was done for VICReg. We apply the regularization on representations learned by the encoder, the same as in LDReg. The results can be found in the table below. All results are based on 100 epoch pretraining with BYOL and ResNet-50 on ImageNet. All experimental settings are exactly the same as LDReg except for the regularization term.
>
> | Method | Regularizer | $\beta$ | Linear Evaluation Accuracy | Effective Rank | Geometric mean of LID |
> |:------:|:-----------:|:-------:|:-----------------:|:--------------:|:---------------------:|
> | BYOL |     None    |    -    |        67.6       |      583.8     |          15.9         |
> | BYOL |      Cov     |   0.01  |        67.6       |      583.5     |          15.9         |
> | BYOL |      Cov     |   0.1   |        67.5       |      593.5     |          15.8         |
> | BYOL |   Cov+Var   |   0.01  |        67.8       |      539.2     |          15.5         |
> | BYOL |   Cov+Var   |   0.1   |        67.7       |      798.4     |          16.8         |
> | BYOL |    LDReg    |  0.005  |      **68.5**     |      594.0     |          22.3         |
>
> Although the covariance with variance (Cov+Var) regularizer can increase the global dimension, it does not improve the local dimension. It also has a very minor effect on the linear evaluation accuracy, whereas LDReg improves by almost 1%. This further supports the benefit of LDReg, focusing at the local rather than global level.

---

### Meta-Review · Area_Chair_FBUR · 2023-12-07

**Metareview:**

Thanks for your submission to ICLR.

This paper examines regularizing the LID of features to improve performance of deep networks.  Four reviewers took a look at this paper.  Strengths included that the paper is well-written, there is good theory and theoretical underpinnings, and that this is an important problem.  Some of the noted weaknesses included several (minor) issues with experiments, and a question about computational cost of the method.

The authors did a nice job responding to the reviews.  Indeed, multiple reviewers said that they would raise their score.  Ultimately, the scores indicate that three of the four reviewers are on the accept side (8, 6, 6) and one is on the reject side (5); however, the reviewer who gave a 5 also indicated that they were happy with the rebuttal and was willing to raise their score.  As far as I can see, at the end of the day all four reviewers seem happy with the paper and are advocating for its acceptance.

**Justification For Why Not Higher Score:**

This could potentially be moved to a spotlight, but in general the scores were on the borderline side so I'm not confident enough to argue for making this a spotlight paper.

**Justification For Why Not Lower Score:**

Ultimately all of the reviewers are positive about the paper and felt that the rebuttal answered lingering questions.  None of the reviewers are advocating to reject this paper.

---

### Decision · Program_Chairs · 2024-01-16

Accept (poster)